# "The future depends on what we do in the present" - Development positions of EU countries by levels of sustainable development and living standards

Aleksandra Łuczak[1], Klara Cermakova[2]*, Sławomir Kalinowski[3], Eduard Hromada[4], Oskar Szczygieł[5]

1 Department of Finance and Accounting, Faculty of Economics, Poznań University of Life Sciences, Poznań, Poland, 2 Department of Economics, Faculty of Economics, Prague University of Economics and Business, Prague, Czechia, 3 Department of Rural Economics, Institute of Rural and Agricultural Development, Polish Academy of Sciences and Committee on Labour and Social Policy Sciences, Polish Academy of Sciences (Komitet Nauk o Pracy i Polityce Społecznej PAN), Warsaw, Poland, 4 Department of Construction Management and Economics, Czech Technical University in Prague, Prague, Czechia, 5 Department of Rural Economics, Institute of Rural and Agricultural Development, Polish Academy of Sciences, Warsaw, Poland.

* klara.cermakova@vse.cz

## Abstract

Sustainable development and standard of living of households have recently become central topics in economic analysis, scientific research, and public debate. This study aims to evaluate the positions of European Union (EU) countries and the changes occurring between 2016 and 2023 in terms of sustainable development and living standard. A hybrid multi-criteria decision-making process based on the modified positional technique for order of preference by similarity to the ideal solution (MP-TOPSIS) was employed to assess these dimensions. The research utilized data from the Eurostat database for the specified period. Both the standard of living of EU inhabitants and their countries' levels of sustainable development were assessed. The findings revealed a clear relationship between the households' living standards and the degree of sustainable development in EU countries. Furthermore, the analysis highlighted significant differentiation in these levels across countries, alongside a general upward trend throughout the period analyzed.

## Introduction

Reflections on the evolution of territorial units and standard of living have been ongoing for a considerable time. A straightforward definition of development as transformation persisted until the 18th century. However, Kudarewska [1] points out that the secular philosophical concept of development emerged as early as the 17th century and signified that "all societies move forward in a natural and continuous way on the

**Data availability statement:** All relevant data are within the manuscript and its Supporting Information files (Variables.xlsx).

**Funding:** This study was funded by the Prague University of Economics and Business in the form of a grant [IGA F5/2023/4 to KC]. The Prague University of Economics and Business also provided support in the form of funding for APC coverage. The funders had no role in study design, data collection and analysis, decision to publish, or preparation of the manuscript.

**Competing interests:** No competing interests.

way from poverty, barbarism, despotism and ignorance to wealth, civilization, democracy and rationality" [1]. In the 18th and 19th centuries, definitions of development found their foundation in rationalism and humanism. With the advent of the Industrial Revolution, the notion of development acquired an economic nature. Nevertheless, by the close of the 19th century, the concepts of development and progress had become distinctly separate. Development came to be associated with Christian order, responsibility, and modernization, while progress took on a character of disorder, which was observed "in the rapid industrialization of some regions of Western Europe and North America" [2].

Against the backdrop of these general explanations of development, we can delve into more specific and empowering definitions. Assigning development, a tangible "shape" involves defining it within a specific area; in such a context, it becomes territorial development. Brol [3] claims that the notion of territorial development lacks a single unambiguous definition and "is a mental shortcut covering a wide spectrum of economic, social, and spatial processes taking place in a specific territory."

Development is marked by purposiveness, awareness, a long-term horizon, social and local dimensions, and interventions carried out by governmental authorities. Purposiveness encompasses all actions aimed at achieving a clearly defined objective or outcome [4]. Broński [5] emphasizes that the primary aim of local development is to enhance the quality of life of the local population. Links between quality of life and sustainable development are also explored Costanza et al. [6]. Schumpeter's [7] definition diverges from classical definitions by presenting development as a shift from one state in the economic system to another, in a manner not reducible to incremental smaller steps. He describes development as a non-continuous transformation, evident both in the intensity of the phenomenon and the disruption of the current equilibrium, ultimately leading to a new trend.

In the context of this paper, development refers to positive transformations. Our research seeks to evaluate the level of development while simultaneously tracking the changes occurring over time (thus incorporating a longer perspective). Broadly speaking, development at regional and local levels is predominantly socio-economic in nature. Presently, territorial development integrates economic and social activities while also prioritizing environmental protection. In this view, socio-economic development aligns with sustainable development, which is now the guiding principle in local development research.

The concept of sustainable development remains highly pertinent in the 21st century. It was initially introduced in 1980 within the documents of the "World Nature Conservation Strategy," and was subsequently elaborated in the World Commission on Environment and Development report, often referred to as the Brundtland Report of 1987 – "Our Common Future" [8]. This report asserts that sustainable development is both achievable and necessary to fulfill the needs of present generations without compromising the ability of future generations to meet their own needs. Wilkin [9] observed that "the concept of sustainable development has become the paradigm of contemporary development theories," with numerous territorial units adopting it as a framework for planning their development.

Sustainable development should incorporate the needs and participation of residents within territorial units. A prerequisite for sustainable development is the awareness of local communities regarding their situation across various dimensions, enabling them to identify and address emerging challenges. Consequently, it is essential to examine sustainable development alongside the standard of living of residents. Emphasis must be placed on ensuring sustainable development leads to increased regional economic potential, enhanced competitiveness, and, above all, improved standards of living for inhabitants.

In the context of the dynamic changes taking place in Europe, as well as global challenges such as climate change, the COVID-19 pandemic, and geopolitical conflicts, the analysis of the standard of living and sustainable development in EU countries is of particular importance. Studies conducted by experts such as Stiglitz, Sen, and Fitoussi [10] emphasize the need to consider a wide range of indicators that go beyond traditional economic measures such as GDP. In their report "Mismeasuring Our Lives," these authors argue that social progress should be measured using indicators that reflect quality of life, well-being, equality, and sustainable development. Therefore, studying the standard of living in the context of sustainable development requires considering both economic and social and environmental aspects [11,12].

Additionally, it is worth paying attention to the concept of "inclusive growth," which is gaining increasing recognition in the context of development policy [13,14]. This growth, as defined by the OECD, should not only be sustainable but also generate opportunities for all social groups, reducing inequalities and promoting social cohesion. In the context of sustainable development, inclusive growth means that the benefits of development should be shared fairly, and the burdens associated with the ecological transformation should not disproportionately affect marginalized groups [15]. The study of the standard of living in the context of sustainable development should therefore also take into account aspects related to income distribution, access to education and healthcare, as well as the possibility of participation in social and political life [16,17]

An increasingly prominent concept in discussions of economic growth is inclusive growth, recognized as a crucial factor for reducing poverty [18]. Sustainable development, as well as sustained and inclusive growth, is embedded within Goal 8 of the United Nations global goals and is being implemented across EU countries with a target date of 2030 [19].

The concept of sustainable development has become a central pillar of contemporary economic discussions, signalling a heightened awareness of the necessity to balance economic growth with social equity and environmental stewardship. Within the European Union (EU), the pursuit of sustainable development is closely tied to the standard of living of its residents, encompassing aspects like quality of life, social well-being, and economic prosperity. EU economic policies are structured to align with sustainability objectives across real-life scenarios [20]. The period from 2017 to 2023 was a pivotal phase in the EU's development trajectory, characterized by substantial shifts in economic policies, social trends, and environmental strategies, significantly shaped by the COVID-19 pandemic. As noted by Jasova [21] and Zubikova [22] the pandemic ushered in new perspectives on sustainability and sustainable development that had not been previously considered.

Several aspects of sustainable development, as highlighted within the literature, underscore its critical importance in scientific discourse. From the vantage point of this research, attention should be directed to the sustainability of territorial units and their developmental transformations, particularly in the context of evolving standards of living. Why is this significant? Because these changes enable assessments of socio-economic progress among residents of various regions and the circumstances within their respective regions or countries. Development disparities and a lack of regional convergence were explored, for instance, by Řežábek, et al. [23] in their comparative analysis of regional development within V4 countries. This study integrates two critical concepts: the standard of living, reflecting prosperity and access to fundamental resources, and sustainable development, advocating for balanced advancement across social, economic, and environmental dimensions [24,25].

The foundation of this discussion is rooted in the contributions of distinguished authors such as Amartya Sen [11,26] and Jeffrey Sachs [27,28]. These scholars have profoundly influenced our understanding of sustainable development

                                                                                    

and the standard of living, each presenting a distinctive viewpoint on the intricate factors influencing human well-being. Amartya Sen, a Nobel laureate in economics, introduced the concept of "development as freedom," emphasizing that individual well-being should be assessed based on the ability to lead lives they value. Sen's framework considers not only material prosperity but also elements such as freedom of choice, access to education, healthcare, and political participation. This perspective broadens the assessment of well-being beyond conventional measures like GDP and income, recognizing human development as a multi-dimensional pursuit. In contrast, Jeffrey Sachs, recognized for his dedication to the Sustainable Development Goals (SDGs), has played a pivotal role in advancing poverty alleviation and enhancing the quality of life in developing nations. Sachs underlines the significance of international collaboration and political efforts in combating global issues such as poverty, climate change, and social inequities. His work provides a comprehensive strategy for improving the standard of living on a global scale.

Equally valuable is the exploration of sustainable development through the lenses of Ostrom, [29] and Raworth [16]. Ostrom's research concentrated on the sustainable management of common resources such as forests and pastures. Her findings illustrate how local communities can successfully oversee resources while supporting sustainable development, an essential aspect of global ecological balance. Raworth, on the other hand, introduced the concept of "Doughnut Economics," which links sustainable development objectives with the planet's ecological boundaries. Her model advocates for an economic system that operates within the "doughnut," avoiding both inadequate societal well-being and the surpassing of ecological limits. These perspectives underscore the risks of improving economic conditions without considering sustainable development, as such actions jeopardize our planet's ecological boundaries and the future well-being of generations.

These viewpoints align closely with the work of Daly [30], who, in his book "Beyond Growth..." (1996), argued for transitioning from the relentless pursuit of economic growth to an economic system centred on sustainable resource usage. Daly's insights draw attention to the finite nature of planetary resources and the environmental repercussions of unrestrained growth. His perspective heightens the urgency to reimagine traditional economic approaches in favour of models that prioritize ecological sustainability. By integrating such ideas, we can work toward achieving an equilibrium that ensures human well-being while fostering sustainable development.

The title of this research paper succinctly conveys its central objective, which is to examine the intricate relationship between sustainable development and the standard of living within the EU, while evaluating the status and developmental shifts of individual member states in these areas. The study sought to evaluate the positions of EU countries and their changes concerning sustainable development and the standard of living of their inhabitants. This research addresses a gap in the literature by linking sustainable development to the standard of living in its analysis. It proposes a holistic framework for investigating these two interconnected issues and applies it to the context of EU countries.

The comprehensive procedure introduced in this study is referred to as an integrated procedure. This methodology combines elements of sustainable development and standard of living analyses, offering a more effective approach to examining the situation of countries striving towards sustainable development goals and a high standard of living for their residents than separate analyses of these topics would provide. To conduct this analysis, a hybrid multi-criteria decision-making procedure was employed for evaluating the level of sustainable development and the standard of living in EU countries. Specifically, the procedure is based on the modified positional technique for order of preference by similarity to the ideal solution (MP-TOPSIS). The research utilized data extracted from the Eurostat database for the period 2016–2023.

Building on existing research gaps, the following research questions were posed:

1. What were the levels of sustainable development and standard of living in European countries during 2016–2023?

2. What were the developmental positions of countries during the studied period?

3. What changes occurred in the development positions of EU countries between 2016 and 2023?

4. Which factors related to the standard of living most significantly influence sustainable development, and vice versa?

We formulated these hypotheses in our research.

Hypothesis 1: The levels of sustainable development and living standards in European countries from 2017 to 2023 showed significant differences, with Northern European countries achieving higher indicators in both areas compared to Southern and Eastern European countries.

Hypothesis 2: The developmental positions of European countries with high levels of sustainable development and living standards were stable during the studied period, with minor changes in the rankings.

Hypothesis 3: European countries with low levels of sustainable development and living standards experienced significant changes during the studied period.

Hypothesis 4: Higher average income positively influences sustainable development, while deprivation and financial difficulties hinder sustainable development.

Hypothesis 5: Higher rates of organic farming and recycling positively influence the standard of living, while higher risk of poverty and social exclusion hinder improvements in the standard of living.

The comprehensive approach proposed in the study expands the analytical possibilities in exploring sustainable development and the standard of living at the territorial level. The results of this research have practical implications, as they can guide the development planning of territorial units and support the formulation of coherent social policies.

The article, in addition to the introduction, contains five more sections. the rest of the paper is organized as follows: literature review, data and methods, empirical results and discussion, and conclusions, and implications and recommendations.

## Literature review

The challenges of sustainable development facing the modern world – such as poverty, social inequalities, climate issues, and resource limitations – demand not only countermeasures but also thorough analyses. The implementation of the Sustainable Development Goals (SDGs) offers a potential response to these challenges, addressing economic, environmental, and social dimensions with a long-term perspective. However, their successful implementation relies on the prior diagnosis of the existing situation and appropriate development programming.

The integration of sustainable development into the strategies of countries, regions, and local units has necessitated the creation of methods to measure it. While the literature offers numerous approaches for separately examining both sustainable development and the standard of living of residents, there is a notable lack of research addressing these two dimensions simultaneously. Various methods and tools are employed in studies assessing sustainable development (cf. [25,31–35]). It is important to highlight that the evaluation of sustainable development must rely on a diverse array of multidimensional indicators [36]. Furthermore, "the assessment of sustainable development is usually based on quantitative indicators, and becomes more complex the more indicators are considered" [37].

A comprehensive examination of sustainable development can be found in the work of Łuczak and Just [38], which demonstrated the potential application of a multi-criteria decision-making approach, incorporating optimal tail selection, in assessing the sustainable development of territorial units. This research highlights the need for innovative approaches capable of addressing the intricacies of measuring sustainable development in an integrated and thorough manner.

The publication by Pereverzieva and Volkov [39] The study is notable for its proposal of a method for calculating an integrated development index for EU member states for the period 2016–2017, incorporating scenario forecasting up to 2020. The study integrated indicators such as the human development index, life satisfaction, and environmental performance index. It tested the hypothesis that a high-income level does not necessarily equate to high life satisfaction in a country. This approach identifies countries with a balanced development of their economic, human, and ecological systems. Further studies, such as those by Setek and Kopp et al. [40,41], explored the importance of addressing circular economy and water scarcity and efficient water resource management.

Another publication with a national focus, authored by Filiz Karafak [42], analyzed Finland's integrated water management as an example of the EU's hydro-political approach to sustainable ecological planning. Finland, rich in water resources, has developed diverse policies ranging from sustainable use to wastewater reclamation, contributing to resource preservation. The study sheds light on EU hydro-political strategies and offers valuable insights into sustainable water management practices.

On an urban scale, sustainability research was conducted by Przybylowski [43], who focused on the city of Gdynia as part of the CIVITAS DYN@MO European projectPrzybyłowski. The study examined challenges in sustainable urban mobility planning, emphasizing stakeholder participation, integrated ticketing systems, parking management, and pedestrian inclusion. It offered a comprehensive perspective on urban development challenges related to mobility.

Another dimension of sustainability was explored by Ebener et al. [44], who proposed standardized geographical indicators for emergency obstetric and newborn care in low- and middle-income countries. Though not directly linked to the EU, the study highlighted global healthcare system performance and its connection to sustainable development. It stressed the significance of physical accessibility and high-quality emergency care, contributing to the broader discourse on sustainability.

A rural perspective was presented by Palapa, Toniuk, Nagorniuk, and Hutsol [45], who examined sustainable development in Ukrainian rural areas through the lens of developed countries, including EU practices. The study identified core challenges and advancements in rural development in Ukraine while addressing ecological and social aspects, offering insights into rural sustainability practices. In a study by Seto et al. [46], the relationship of sustainability in rural and local contexts is noted, as well as the impact of urbanization on the environment and biodiversity.

Sustainable development is fundamentally about balancing economic growth, social inclusion, and environmental protection. Bebbington and Unerman highlight the evolution of research into sustainable development, emphasizing the need for integrated approaches that consider the interdependencies between these dimensions [47]. This is echoed by Burlacu et al. [48], who argue that digital public administration can enhance sustainable development by improving decision-making processes and promoting transparency [48]. The role of public administration is crucial, as effective governance structures can facilitate the implementation of sustainable practices and policies that directly affect living standards [49].

The findings from various studies indicate a significant correlation between the levels of sustainable development and the standard of living across EU countries. For instance, Kibet et al. emphasize that public participation in development programs is essential for achieving sustainable outcomes, as it ensures that the needs and perspectives of local communities are integrated into decision-making processes [50]. This participatory approach is vital for fostering a sense of ownership and accountability among citizens, which can lead to improved living standards and sustainable urban development [51].

Moreover, the analysis reveals considerable disparities in sustainable development and living standards among EU member states. Research by Zysk highlights that urban management policies must address local community needs and incorporate renewable energy technologies to enhance sustainability [51]. This is particularly relevant in the context of the COVID-19 pandemic, which has exacerbated existing inequalities and highlighted the need for resilient urban planning that prioritizes sustainability [52–56]. The work of Grover further illustrates how public support for sustainable initiatives, such as higher density living and public transit, has fluctuated, indicating a need for renewed advocacy for sustainable urban policies [57].

The upward trend in sustainable development indicators from 2016 to 2023 suggests a growing commitment among EU countries to integrate sustainability into their economic frameworks. Meschede and Henkel argue that the intersection of library and information science with sustainable development can provide valuable insights into community engagement and knowledge dissemination [58]. This is crucial for fostering a culture of sustainability that permeates various sectors of society, ultimately contributing to improved living standards.

Other research, such as that by Kryshtanovych et al. [59–61], evaluated the circular economy in 20 EU countries by focusing on minimizing environmental consequences and economic costs. Their publication examined the progress and principles behind the circular economy model, creating tools to analyze processing capabilities and contributing to the understanding of sustainable resource management.

Kaklauskas et al. [62] tackled trends and progress in the sustainable construction industries across the EU, the UK, and Norway. Using methods such as COmplex PRoportional Assessment (COPRAS) and Degree of Project Utility and Investment Value Assessments (INVAR), the study provided valuable insights to consumers for rational decision-making based on country-specific sustainable construction performance.

Housing sustainability was explored by Napiórkowska-Baryła and Świdyńska [63], who examined housing conditions in Cittaslow towns in Poland. The study analyzed factors influencing housing quality and affordability, contributing to the understanding of sustainable urban housing markets and offering localized insights into housing sustainability practices.

Research tied to water and wastewater management in the sustainability context was presented by Łakomy-Zinowik [64], who investigated public-private partnerships (PPPs) and their role in fostering sustainable water management. By analyzing practical examples, the study highlighted the potential benefits and challenges of collaboration between public and private sectors in achieving sustainable water resource goals.

In the energy sector, the study by Matuszewska-Janica et al. [65] focused on energy consumption patterns and renewable energy usage among EU households, providing guidance for policymakers and energy providers to enhance sustainable energy practices. Bednar [66] examined the macroeconomic implications of renewable energy adoption, showing that countries with a higher share of renewable energy tend to experience lower inflation rates. On the micro level, Cermakova [67] highlighted how energy and housing affordability influence household poverty across the EU.

Additionally, Momete [68] analyzed the progress of European countries toward energy efficiency, offering critical insights into the successes and obstacles encountered by EU member states in energy planning and implementation. Similarly, Rybak et al. [69] evaluated the integration of EU-27 energy markets in the context of sustainable development and SDG7 implementation. Their study highlighted the challenges, opportunities, and progress made in energy market integration, contributing to the understanding of sustainable energy practices within the European Union.

Turning to research closely related to sustainable development and the standard of living, the work of Rajnoha, Lesníková, and Vahančík [70] deserves particular attention. They investigated the relationship between sustainable development and quality of life in V4 countries and Austria through an evaluation of 26 quality-of-life indicators. Their research highlighted the positive influence of GDP growth on 61.5% of the variables studied while revealing a significant sustainable development gap between the V4 countries and Austria. In this regards Jasova et al [71] highlights dichotomy in economic development and asymmetrical processes on labor markets under changing institutional, economic and legal framework.

An innovative approach to examining sustainable development goals within the context of health and well-being was introduced by Sompolska-Rzechuła and Kurdyś-Kujawska [72]. Their study analyzed the implementation of climate action and good health and well-being goals in EU countries, focusing on trade-offs and synergies between these goals by employing the TOPSIS method. The findings emphasized the varying degrees of goal implementation and the vital role of energy policy in advancing these objectives, contributing to a deeper understanding of the interactions between sustainable development goals.

Another notable publication is that of Misztal [73], who analyzed sustainable development and quality of life in Poland compared to other OECD member states. The study, spanning the years 2000–2018, revealed tangible progress in social and individual well-being in Poland, including substantial improvements in personal safety, education, and work-life balance. However, the country was shown to lag behind the OECD average in other areas, offering a nuanced perspective on Poland's sustainable development progress. Rotschedl et al [74] highlight the importance of sustainable pension system as resonant aspect of sustainable development especially within the EU countries.

The integration of immigrants, an issue intersecting with several SDGs, was the focus of research by Zubíková [22,75]. Her findings underscored stark disparities in the level of immigrant integration across EU member states, revealing that immigrants face a higher risk of poverty compared to the native population. Vokoun et al [76] highlight the relationship between sustainable development and level of criminality. Their study brings evidence about synergies between economic development and various aspects of standart of living.

The relationship between sustainable development and quality of life was also explored by Gryshova et al. [77], who evaluated how industrial structure impacts sustainable economic development and quality of life, including within the EU. Using cluster analysis and integral assessment, they confirmed the hypothesis that progressive industrial structures positively influence sustainability and quality of life. Similar results are achieved by Popescu [78] who offs a comprehensive exploration of the dynamic interplay between changing economic environment and sustainable development.

Tourism, quality of life, and sustainability were linked in the work of Băndoi et al. [33], who assessed the relationship between tourism development and quality of life in the EU. Their research, which utilized cluster analysis, found a positive correlation between tourism intensity, sustainable performance, and quality of life. The study suggested opportunities for future research into causal relationships and the development of cohesive public policies to support sustainable growth.

Amidst this research, the impact of the COVID-19 pandemic on sustainable development received prominent attention in the 2021 Eurostat report, which evaluated the EU's progress towards achieving SDGs. While the report highlighted advancements in poverty reduction and health, it also detailed setbacks due to the pandemic's effects on the economy and labor market. Similarly, Lafortune et al. [79] underscored the pandemic's adverse impact on sustainability in Europe while emphasizing the region's leadership role in prioritizing SDGs globally. They stressed the necessity of maintaining focus on areas such as sustainable diets, agriculture, and biodiversity.

The Eurostat 2023 report continued this thread by addressing how recent crises, including the energy crisis and the pandemic aftermath, underline the urgency of accelerating SDG-related efforts. Meanwhile, De Neve and Sachs [80] explored the interplay between SDGs and subjective well-being, identifying strong correlations but also complexities, such as negative associations between some environmental goals and well-being. This calls for more nuanced policy approaches.

Within this broader context of evolving understandings of sustainability in economic development and growth, our research will undertake a comprehensive, multi-criteria analysis of the intricate relationship between sustainable development and the standard of living within the European Union. Our analysis focuses on the interconnected dimensions of economic prosperity, social well-being, and environmental sustainability, aiming to provide new insights into this multi-faceted relationship.

## Data and methods

The research is based on Eurostat data on sustainable development and living standards for European Union countries from 2017 to 2023. This comprehensive dataset provides a robust foundation for analyzing various socio-economic indicators across EU member states. By leveraging this data, the study aims to offer insights into trends and patterns in development, allowing for a detailed comparison of progress among different countries. The analysis also incorporates scenario forecasting, which helps to project potential future developments and assess the impact of various policy measures. This approach ensures that the findings are not only reflective of past and present conditions but also provide a forward-looking perspective on the potential trajectories of EU member states.

During the period 2017–2023, the European Union consisted of 27 member states. The research encompasses all EU member states during the period 2017–2023. Each of these countries brought unique characteristics and diverse experiences to the community, contributing to the cultural and economic richness of the EU. Countries like Sweden and Finland were known for their advanced technologies, innovative industries, and high standards of living, which set benchmarks for other nations. Southern countries such as Greece and Portugal stood out for their rich historical heritage, vibrant cultures, and thriving tourism sectors, attracting millions of visitors each year and significantly boosting their economies.

Eastern member states, such as Poland and Romania, were dynamically developing their economies, focusing on modernization and infrastructure improvements. These countries were striving for sustainable development, implementing policies aimed at reducing environmental impact while fostering economic growth. Additionally, they were committed to improving the living standards of their citizens through investments in education, healthcare, and social services.

Western European countries, including Germany and France, continued to play pivotal roles in the EU's political and economic landscape. Germany, with its strong industrial base and robust economy, was a driving force behind many EU initiatives. France, known for its cultural influence and agricultural production, also contributed significantly to the EU's overall prosperity. Overall, the diverse contributions of these member states enriched the European Union, making it a unique and multifaceted entity on the global stage.

Fig 1 outlines an eight-step process to evaluate changes in the development positions of EU countries, focusing on sustainable development and living standards from 2017 to 2023.

The process begins by creating data matrices for sustainable development and living standards indicators. Variables are classified as either benefit or cost variables, with cost variables converted into benefit variables for consistency. Normalization follows to standardize the data, employing winsorization to address outliers and positional techniques for asymmetrical distributions.

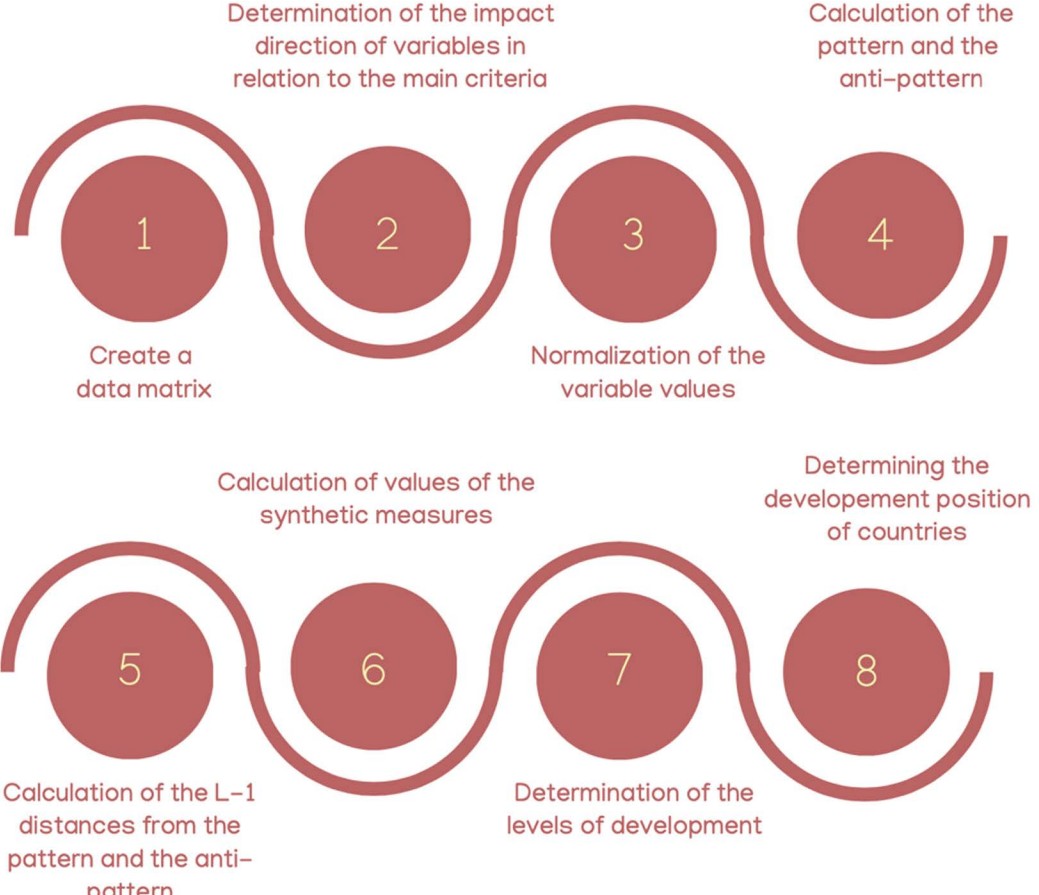

**Fig 1. Procedure for assessment of changes in the development position of European Union countries.** *Source: Own elaboration based on Łuczak (2016).*

Patterns (ideal points) and anti-patterns (anti-ideal points) are then calculated, serving as benchmarks for evaluation. Distances between countries' indicators and these benchmarks are measured using the L1 metric, forming the basis for synthetic measures. These measures, ranging from 0 to 1, provide a comprehensive assessment of each country's performance.

Countries are subsequently classified into six development levels and four main development positions: strong effective, preemptive, moderate, and weak. This framework highlights the interconnected dynamics of sustainable development and living standards, supporting detailed analysis and policy recommendations.

In this research, we employed a procedure based on the modified positional technique for order of preference by similarity to the ideal solution (MP-TOPSIS) to assess the development positions of European Union countries. TOPSIS, introduced by Hwang and Yoon [81], remains one of the most recognized and widely used multi-criteria decision-making methods due to its simplicity and adaptability. In this study, the method was utilized for the construction of a two-pattern synthetic measure, allowing for an integrated evaluation of sustainable development and living standards.

The concept of constructing a synthetic measure was originally pioneered by Hellwig [82] as a single-pattern approach. Over the years, TOPSIS has undergone significant enhancements, with numerous modifications tailored to address diverse research challenges and datasets. It is now applied across various domains, including sustainability analysis, resource management, and socio-economic studies (e.g., [83–87]).

The procedure used in this study systematically follows eight distinct steps, as illustrated in Fig 1, encompassing data pre-processing, normalization, calculation of reference points, and classification of development levels. This structured approach ensures a robust and objective analysis, making it well-suited for evaluating complex phenomena like sustainable development and standard of living dynamics.

In the first step, panel data for territorial units is organized into a data matrix encompassing $K$ variables for $N$ countries over $T$ years. Specifically, two separate data matrices are constructed: one representing indicators of sustainable development and the other reflecting variables related to the standard of living. These matrices capture the socio-economic and environmental dynamics of European Union countries during the period 2017–2023. This structured data organization forms the basis for a comprehensive analysis of development trends and facilitates subsequent computations of synthetic measures:

$$\mathbf{X} = \begin{bmatrix} x_{111} & x_{121} & \dots & x_{1K_1 1} \\ x_{211} & x_{221} & \dots & x_{2K_1 1} \\ \dots & \dots & \dots & \dots \\ x_{N_1 1T} & x_{N_1 2T} & \dots & x_{N_1 K_1 T} \end{bmatrix} \tag{1}$$

$$\mathbf{Y} = \begin{bmatrix} y_{111} & y_{121} & \dots & y_{1K_2 1} \\ y_{211} & y_{221} & \dots & y_{2K_2 1} \\ \dots & \dots & \dots & \dots \\ y_{N_2 1T} & y_{N_2 2T} & \dots & y_{N_2 K_2 T} \end{bmatrix} \tag{2}$$

where: $\mathbf{X}$ – data matrix for sustainable development, $\mathbf{Y}$–data matrix for the standard of living, $x_{ijt}$ ($y_{ijt}$)– value of $j$-th ($j = 1, 2, \dots K_1$ ($K_2$)) variable for sustainable development (standard of living) for $i$-th country ($i = 1, 2, \dots N$) in $t$-th year ($t = 1, 2, \dots, T$), $K_1$ ($K_2$) – number of variables describing sustainable development (standard of living), $N$ – number of countries, $T$ – number of years analysed.

In the second step, the direction of impact for each variable in relation to the main criteria is determined. Variables are classified as either benefit variables or cost variables. Benefit variables positively contribute to increasing the phenomenon being studied, while cost variables negatively influence it. To ensure consistency in the analysis, cost variables are

transformed into benefit variables using a differential transform. Additionally, since variables are often expressed in different units and ranges of values, they must be unified into stimulants during the normalization process to allow for consistent comparisons across all variables.

In step three, the normalization of variable values is performed. Numerous standardization techniques are discussed in the literature (e.g., [88,89]), and the choice of the appropriate method depends on the type and distribution of the analysed data [90]. Before normalization, it is essential to examine the distribution of each variable. Based on recommendations by Łuczak and Just [38,91] different normalization techniques are applied depending on the nature of the distribution.

First, if all variables exhibit a normal or near-normal distribution, classical standardization techniques are employed. Second, variables showing asymmetry without extreme values are normalized using standardization based on the spatial median, which reduces the influence of skewness. Third, if extreme values are present, the data should undergo winsorization to limit their impact. Finally, for variables with very strong asymmetry combined with extreme observations, a combination of winsorization and positional normalization is applied. These tailored approaches ensure the robustness and reliability of the normalized datasets, paving the way for accurate analysis in subsequent steps.

Panel data for territorial units frequently include extreme observations and exhibit asymmetric distributions, which can skew the analysis. To address this, a two-step process is applied. First, winsorization is used to reduce the influence of outliers by capping extreme values at predefined thresholds. This ensures that data distributions are less affected by anomalies. Second, normalization is performed using the spatial median, a robust method that minimizes the impact of asymmetry and provides more reliable standardized values. This approach enhances the comparability of variables across different units and ensures the integrity of the analysis. Winsorization consists in descending extreme values of variables with constants – thresholds which are determined on the quartile criterion [92] as follows:

$$\left(\textit{lower limit}_j, \textit{uper limit}_j\right) = \left(ll_j, \ ul_j\right) = \left(Q_{1j} - 1.5 \cdot IQR_j, \ Q_{3j} + 1.5 \cdot IQR_i \right) \tag{3}$$

where: $ll_j$, $ul_j$—respectively lower and upper limits $j$-th variable, to which occur extreme values, $Q_{1j}$, $Q_{3j}$—respectively the first quartile (25th percentile) and third quartile (75th percentile) of the $j$-th variable, $IQR_j$—interquartile range of the $j$-th variable (the difference between the third and first quartiles). The values of the $j$-th variable are collectively analysed over the span of $T$ years. Normalization is a crucial step in the TOPSIS (Technique for Order of Preference by Similarity to Ideal Solution) method because it allows for the transformation of input data into comparable units. In TOPSIS, normalization is essential to ensure that all criteria have an equal impact on the final result, regardless of their original units of measurement. Following winsorization, positional normalization, which is a modified form of standardization based on the spatial median, is applied Following winsorization, positional normalization, which is a modified form of standardization based on the spatial median, is applied (see [93–95]):

$$z_{ijt} = \frac{x_{ijt} - \widetilde{med}_j(x)}{1.4826 \cdot med_i \left|x_{ijt} - \widetilde{med}_j\right|} \tag{4}$$

$$q_{ijt} = \frac{y_{ijt} - \widetilde{med}_j(y)}{1.4826 \cdot med_i \left|y_{ijt} - \widetilde{med}_j\right|} \tag{5}$$

where: $z_{ijt}$ $(q_{ijt})$– normalized value of $j$-th variable ($j = 1, 2, \dots K_1$ $(K_2)$)for sustainable development (standard of living) for $i$-th country ($i = 1, 2, \dots N$) in $t$-th year ($t = 1, 2, \ \dots, \ T$), $\left(\widetilde{med}_1(x), \ \dots, \widetilde{med}_{K_1}(x)\right)\left(\left[\widetilde{med}_1(y), \ \dots, \widetilde{med}_{K_2}(y)\right]\right)$—spatial median vectors of $K_1$ $(K_2)$ variables, 1.4826—constant scaling coefficient (see [94,96])). Proposals for calculating the spatial median were presented by Weber [97–100] a, as well as Kent and Constable [101]. In our study, we calculated the spatial median in the robustX package [102] in the R program [103].

The calculation of patterns and anti-patterns is based on the values of indicators for each criterion. For each criterion, the maximum value is chosen as the pattern and the minimum value as the anti-pattern. In fourth step patterns $(A_1^+, A_2^+)$ and anti-patterns $(A_1^-, A_2^-)$ are calculed as follows for sustainable development:

$$A_1^+ = \left( \max_i (z_{i1t}), \max_i (z_{i2t}), \ldots, \max_i (z_{iK_1t}) \right) = \left( z_1^+, z_2^+, \ldots, z_{K_1}^+ \right) \tag{6}$$

$$A_1^- = \left( \min_i (z_{i1t}), \min_i (z_{i2t}), \ldots, \min_i (z_{iK_1t}) \right) = \left( z_1^-, z_2^1, \ldots, z_{K_1}^- \right) \tag{7}$$

for standard of living:

$$A_2^+ = \left( \max_i (q_{i1t}), \max_i (q_{i2t}), \ldots, \max_i (q_{iK_2t}) \right) = \left( q_1^+, q_2^+, \ldots, q_{K_2}^+ \right) \tag{8}$$

$$A_2^- = \left( \min_i (q_{i1t}), \min_i (q_{i2t}), \ldots, \min_i (q_{iK_2t}) \right) = \left( q_1^-, q_2^-, \ldots, q_{K_2}^- \right) \tag{9}$$

Patterns represent ideal solutions that maximize benefits and minimize costs. Anti-patterns, on the other hand, represent the worst possible solutions that minimize benefits and maximize costs.

Then in the fifth step, the Manhattan distances of each country from the pattern and the anti-pattern are calculated for sustainable development and the standard of living:

$$d_{it}^{1+} = \sum_{j=1}^{K_1} \left| z_{ijt} - z_j^+ \right|, \; d_{it}^{1-} = \sum_{j=1}^{K_1} \left| z_{ijt} - z_j^- \right|, \tag{10}$$

for the standard of living:

$$d_{it}^{2+} = \sum_{j=1}^{K_2} \left| q_{ijt} - q_j^+ \right|, \; d_{it}^{2-} = \sum_{j=1}^{K_2} \left| q_{ijt} - q_j^+ \right| \tag{11}$$

In the sixth step, the distances serve as the foundation for computing synthetic measures using a formula developed by Hwang and Yoon [81] for levels of sustainable development and standard of living:

$$S_{it}^{(\bullet)} = \frac{d_{it}^{(\bullet)-}}{d_{it}^{(\bullet)+} + d_{it}^{(\bullet)-}}, \quad (i = 1, 2, \ldots N, \; t = 1, 2, \ldots, T) \tag{12}$$

where $(\bullet)$ denotes 1 for sustainable development and 2 for standard of living. $S_{it}^{(\bullet)}$ takes values from 0 to 1. The higher the value, the higher the level of development.

The average annual rate of change $r_i$ for jth country of the values of synthetic measures was also calculated:

$$r_i = \frac{-3M + \left[ 9M^2 + 24M(T-1) \left( \frac{1}{S_{i1}^{(\bullet)}} \sum_{t=1}^T S_{it}^{(\bullet)} - T \right) \right]^{1/2}}{2M(T-1)} \cdot 100\% \tag{13}$$

where $M = T(T+1)$. If $r_i > 0$, the studied phenomenon exhibits an upward trend. Conversely, $r_i < 0$ it indicates a downward trend. If $r_i = 0$, it signifies stagnation in the development of the phenomenon, implying no significant change over time.

Calculating the average annual rate of change based on two extreme values ($S_{i1}^{(\bullet)}$, $S_{iT}^{(\bullet)}$) can be done when the studied phenomenon shows a clear developmental trend, either upward or downward $r_i = \left( \sqrt[n-1]{S_{iT}^{(\bullet)}/S_{i1}^{(\bullet)}} - 1 \right) \cdot 100\%$. If the studied phenomenon shows significant deviations from the upward or downward trend (random fluctuations), the average growth rate is calculated based on all the values of the synthetic measure according formula (13).Next in the seventh step, levels of sustainable development and standard of living are determined using an arbitral manner (cf. [104]). We have adopted six levels according to Fig 2.

The final, eighth step involves determining the development position of countries based on their levels of sustainable development and standard of living. In this step, the coordinates ($S_{it}^1, S_{it}^2$) represent each country's position, capturing its performance in both dimensions. These coordinates allow for a comparative evaluation of countries and provide a clear visualization of their development positions (Fig 3).

Depending on whether a country exhibits above-average or below-average performance in sustainable development and standard of living, four main types of development positions are identified:

1. Strong Effective Position: This represents the most advantageous development position, where countries achieve above-average levels in both sustainable development and the standard of living. These countries are typically leaders in aligning socio-economic prosperity with sustainability goals.

2. Pre-emptive Position: In this scenario, a country demonstrates an above-average level of sustainable development but a below-average standard of living for its inhabitants. This indicates that environmental and structural advancements are not yet reflected in broader improvements in living conditions.

3. Moderate Position: Here, countries have an above-average standard of living but fall short in achieving sustainable development. This suggests a focus on immediate socio-economic prosperity at the expense of long-term sustainability objectives.

4. Weak Position: The least favourable development position, where countries score below average in both sustainable development and standard of living, indicating significant challenges across both dimensions.

This classification offers invaluable insights into the relationship between sustainability and living standards, helping policymakers identify areas requiring targeted interventions to achieve balanced and equitable progress.

To identify the variables influencing the levels of sustainable development and the standard of living of the inhabitants of countries (resulting from synthetic measure ($S_{it}^1, S_{it}^2$), an ordered logit models was used:

$$y_{1it}^* = x_{1it}^T \beta_1 + \varepsilon_{1it} \tag{14}$$

$$y_{2it}^* = x_{2it}^T \beta_2 + \varepsilon_{2it} \tag{15}$$

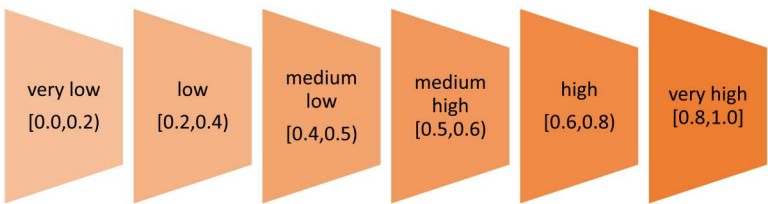

**Fig 2. Levels of development.**

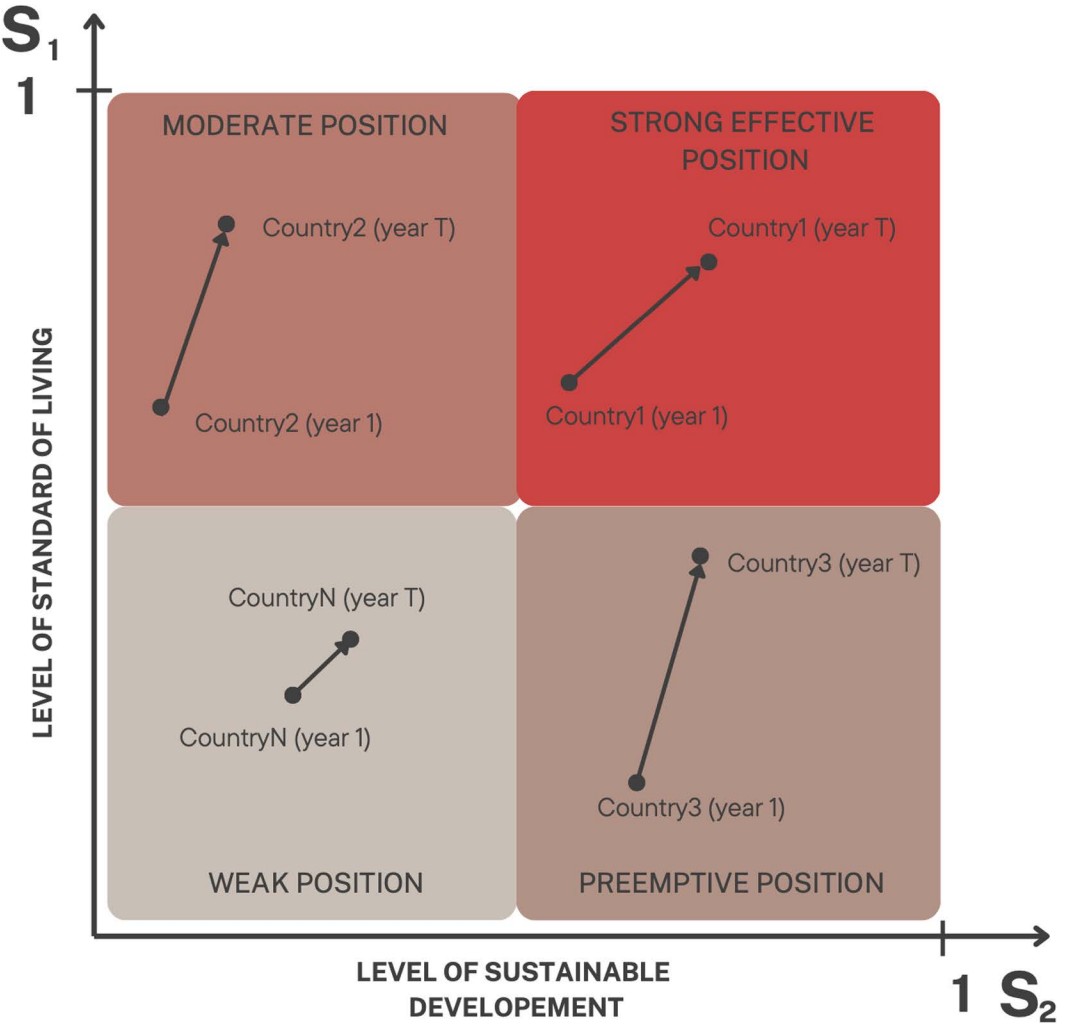

**Fig 3. Types of development positions of countries.** *Source: Own elaboration based on Łuczak (2016).*

where: $y^*_{1it}$, $y^*_{2it}$ – the unobservable variable referring to the *i*-th country in year *t* is related to its discrete counterparts corresponding to levels of sustainable development and standard of living: 6 – very high, 5 – high, 4 – medium-high, 3 – medium-low, 2 – low, 1 – very low, $x^T_{1it}$, $x^T_{2it}$ – vector of explanatory variables for the *i*-th country affecting levels of sustainable development and standard of living in year *t*, $\beta_1$, $\beta_2$ – parameter vectors, $\varepsilon_{1it}$, $\varepsilon_{2it}$ – error terms for *i*-th country in year *t*.

In this case, the cumulative logits, i.e., logarithms of the probability ratios of the *i*-th country belonging to a level not higher than *k*-th, are modelled ($p_{ik}$), and the probability opposite to it (1-$p_{ik}$). The level is determined by − a set of exogenous variables (indicators of sustainable development or standard of living) and the error term. In the case of the kth level (the *k*-th class interval based on the synthetic measure), *k*-1 of the logit equations is obtained:

$$logit\,(p_{ik}) = \ln \frac{P\,(y_i \leq k)}{P\,(y_i > k)} = \beta_0 + \beta_1 x_{1it} + \ldots + \beta_k x_{K_{(\bullet)}it} + \varepsilon_{it}\ for\ k = 1,\ 2,\ldots,K_{(\bullet)}$$

(16)

where: (●) denotes 1 for sustainable development or 2 for standard of living.

The calculated ordered logit models have been computed using the Gretl software [105].

## Empirical results

To assess the position of EU countries and their changes in terms of sustainable development and the standard of living of their inhabitants variables were chosen based on statistical and substantive criteria. The following variables were used to assess the sustainable development: social dimension – people at risk of poverty or social exclusion (%, $x_1$), percentage of the total population aged less than 65 living in households with very low work intensity (%, $x_2$), tertiary educational attainment (%, $x_3$), early leavers from education and training in age group 18–24 (%, $x_4$), proportion of population aged 65 and over (%, $x_5$); economic dimension – real GDP per capita (euro, $x_6$), percentage of the total population of young people neither in employment nor in education and training (NEET) (%, $x_7$), employment rate (%, $x_8$), long-term unemployment rate (%, $x_9$), employment rate of low skilled persons in age group 20–64 (%, $x_{10}$); environmental dimension – final energy consumption in households per capita (kilogram of oil equivalent, $x_{11}$), recycling rate of municipal waste (%, $x_{12}$), average $CO_2$ emissions per km from new passenger cars (g $CO_2$ per km², $x_{13}$), share of renewable energy in gross final energy consumption (%, $x_{14}$), share of area under organic farming in the total utilized agricultural area (%, $x_{15}$).

Standard of living described following variables: mean income of population aged 18 and over (euro, $y_1$), material and social deprivation rate (%, $y_2$), severe material and social deprivation rate (%, $y_3$), people having a long-standing illness or health problem (%, $y_4$), self-reported unmet need for medical examination and care (%, $y_5$), inability to make ends meet (%, $y_6$), share of inactive population in the total population (%, $y_7$), life expectancy (years, $y_8$), share of people with good or very good perceived health (%, $y_9$), inability to face unexpected financial expenses (%, $y_{10}$), arrears (mortgage or rent, utility bills or hire purchase) (%, $y_{11}$).

We identified variables that were characterized by strong skewness ($x_6$, $x_9$, $x_{11}$, $y_2$, $y_3$, $y_5$, $y_6$, $y_{11}$). Moreover, we also identified using the quartile criterion extreme values, that were observed mainly in the right tails of the distribution of variables ($x_1$, $x_2$, $x_4$, $x_6$, $x_7$, $x_9$, $x_{10}$, $x_{11}$, $x_{14}$, $x_{15}$, $y_2$-$y_7$, $y_{10}$, $y_{11}$) and also in the left tails of the distribution of few variables ($x_5$, $x_8$, $x_{10}$, $x_{12}$, $x_{13}$, $y_4$, $y_7$, $y_9$). We used winsorization of data for variables with extreme values. Next, the nature of the variables was identified, which we divided into benefit variables ($x_3$, $x_6$, $x_8$, $x_{12}$, $x_{14}$, $x_{15}$, $y_1$, $y_8$, $y_9$) and cost variables ($x_1$, $x_2$, $x_4$, $x_5$, $x_7$, $x_9$-$x_{11}$, $x_{13}$, $y_2$-$y_7$, $y_{10}$, $y_{11}$). Cost variables were converted into benefit variables using differential transformation. All variables were transformed using modified median standardization. Then, $L_1$-distances are calculated and used to calculate synthetic measures of the levels of sustainable development (Table 1) and measure of standard of living (Table 2). Next, levels of sustainable development (Table 1) and standard of living were identified (Table 2). The best situation in terms of the level of sustainable development was in countries such as Luxembourg, Denmark, and Austria.

The level of development during the studied period was predominantly high or medium-high. Most countries exhibited a medium level of development (covering medium-high and medium-low sub-levels) in both dimensions—sustainable development and the standard of living of residents. The most significant positive changes in the synthetic index of sustainable development were observed in Bulgaria, Romania, Greece, Portugal, and Spain, with an average annual growth rate exceeding 2% but remaining below 3,5%. Despite these improvements, all these countries (except Portugal) maintained one of the weakest overall levels of development during the latter part of the period. Most of them exhibited low or only occasionally medium-low levels of sustainable development, with Italy consistently reporting a low level throughout the study. Conversely, Austria and the Czech Republic experienced a slight negative annual rate of change (−0.7% and −1.4%). In these countries, sustainable development remained high and medium-high, respectively, until 2019 but showed a decline during the pandemic years (2020–2021). Poland, Lithuania, and France also had negative rates of change, but no more than −0.5%

The dynamics of the changes in the standard of living of residents were more pronounced. In countries like Latvia, Lithuania, and Croatia the average annual growth rate ranged between 2% and 3.3%. Yet, these countries exhibited low to medium-low levels of living standards during the study period. On the other hand, countries such as the Netherlands and Luxembourg consistently demonstrated the best performance, maintaining high or very high living standards. Furthermore, Austria, the Czech Republic, Denmark, Germany, Ireland, Malta, and Sweden sustained a high standard of living

throughout the entire period. Conversely, the lowest living standards (low and medium low throughout the study) were recorded in Romania, Latvia, Lithuania, and Greece. A unique case was Slovakia, which observed a negative annual growth rate (−2.0%) in living standards. However, it should be emphasized that the general decline in the standard of living is largely attributed to the adverse effects of the pandemic from 2020 to 2023.

The levels of sustainable development and living standards were crucial in determining the development positions of countries (see Tables 3 and 4). Fig 4 presents the development positions of European Union countries in 2017 and 2023, illustrating four main categories: strong effective position, pre-emptive position, moderate position, and weak position. In 2017, 11 countries achieved a strong effective position, which expanded to 17 countries by 2023 with the inclusion of Slovakia, Portugal, Poland, Malta, Belgium, and Cyprus. Conversely, the pre-emptive position grouped two countries in both 2017 (Lithuania, Slovakia, and Poland) and 2023 (Lithuania). Moderate positions were held by 6 countries in 2017, while in 2023 only three countries, namely Germany, Spain and Italy, maintained this position.

**Table 1. Values of synthetic measures for the level of sustainable development for European Union countries in 2017–2023.**

| $i$ | Country | Synthetic measures | | | | | | | $ri$ | Levels of development | | | | | | |
|---|---|---|---|---|---|---|---|---|---|---|---|---|---|---|---|---|
| | | 2017 | 2018 | 2019 | 2020 | 2021 | 2022 | 2023 | | 2017 | 2018 | 2019 | 2020 | 2021 | 2022 | 2023 |
| 1 | Belgium | 0.484 | 0.497 | 0.516 | 0.480 | 0.487 | 0.517 | 0.539 | 0.9 | ML | ML | MH | ML | ML | MH | MH |
| 2 | Bulgaria | 0.321 | 0.330 | 0.343 | 0.405 | 0.408 | 0.313 | 0.352 | 2.4 | L | L | L | ML | ML | L | L |
| 3 | Czechia | 0.585 | 0.577 | 0.575 | 0.548 | 0.547 | 0.519 | 0.523 | −1.4 | MH | MH | MH | MH | MH | MH | MH |
| 4 | Denmark | 0.599 | 0.615 | 0.614 | 0.611 | 0.609 | 0.611 | 0.610 | 0.4 | MH | H | H | H | H | H | H |
| 5 | Germany | 0.486 | 0.494 | 0.516 | 0.489 | 0.491 | 0.468 | 0.468 | 0.1 | ML | ML | MH | ML | ML | ML | ML |
| 6 | Estonia | 0.520 | 0.549 | 0.554 | 0.550 | 0.551 | 0.485 | 0.510 | 0.5 | MH | MH | MH | MH | MH | ML | MH |
| 7 | Ireland | 0.562 | 0.574 | 0.598 | 0.604 | 0.601 | 0.636 | 0.657 | 1.8 | MH | MH | MH | H | H | H | H |
| 8 | Greece | 0.320 | 0.340 | 0.377 | 0.373 | 0.370 | 0.383 | 0.400 | 3.4 | L | L | L | L | L | L | ML |
| 9 | Spain | 0.360 | 0.366 | 0.381 | 0.406 | 0.406 | 0.401 | 0.421 | 2.1 | L | L | L | ML | ML | ML | ML |
| 10 | France | 0.505 | 0.516 | 0.515 | 0.501 | 0.500 | 0.488 | 0.505 | −0.1 | MH | MH | MH | MH | ML | ML | MH |
| 11 | Croatia | 0.452 | 0.482 | 0.503 | 0.460 | 0.459 | 0.446 | 0.467 | 0.8 | ML | ML | MH | ML | ML | ML | ML |
| 12 | Italy | 0.310 | 0.317 | 0.331 | 0.333 | 0.334 | 0.318 | 0.356 | 1.4 | L | L | L | L | L | L | L |
| 13 | Cyprus | 0.497 | 0.520 | 0.509 | 0.473 | 0.470 | 0.515 | 0.530 | 0.2 | ML | MH | MH | ML | ML | MH | MH |
| 14 | Latvia | 0.484 | 0.510 | 0.521 | 0.517 | 0.520 | 0.492 | 0.498 | 1.1 | ML | MH | MH | MH | MH | ML | ML |
| 15 | Lithuania | 0.537 | 0.540 | 0.537 | 0.541 | 0.537 | 0.513 | 0.503 | −0.4 | MH | MH | MH | MH | MH | MH | MH |
| 16 | Luxembourg | 0.608 | 0.616 | 0.633 | 0.638 | 0.638 | 0.622 | 0.615 | 0.7 | H | H | H | H | H | H | H |
| 17 | Hungary | 0.442 | 0.468 | 0.471 | 0.454 | 0.453 | 0.419 | 0.425 | 0.3 | ML | ML | ML | ML | ML | ML | ML |
| 18 | Malta | 0.466 | 0.477 | 0.476 | 0.513 | 0.511 | 0.524 | 0.542 | 1.8 | ML | ML | ML | MH | MH | MH | MH |
| 19 | Netherlands | 0.579 | 0.592 | 0.619 | 0.619 | 0.617 | 0.613 | 0.625 | 1.3 | MH | MH | H | H | H | H | H |
| 20 | Austria | 0.644 | 0.650 | 0.657 | 0.594 | 0.594 | 0.614 | 0.633 | −0.7 | H | H | H | MH | MH | H | H |
| 21 | Poland | 0.535 | 0.536 | 0.542 | 0.528 | 0.520 | 0.510 | 0.527 | −0.3 | MH | MH | MH | MH | MH | MH | MH |
| 22 | Portugal | 0.436 | 0.462 | 0.492 | 0.490 | 0.492 | 0.510 | 0.510 | 2.6 | ML | ML | ML | ML | ML | MH | MH |
| 23 | Romania | 0.298 | 0.312 | 0.321 | 0.407 | 0.411 | 0.319 | 0.321 | 3.4 | L | L | L | ML | ML | L | L |
| 24 | Slovenia | 0.595 | 0.606 | 0.627 | 0.618 | 0.612 | 0.583 | 0.565 | 0.3 | MH | H | H | H | H | MH | MH |
| 25 | Slovakia | 0.535 | 0.554 | 0.580 | 0.547 | 0.549 | 0.537 | 0.544 | 0.6 | MH | MH | MH | MH | MH | MH | MH |
| 26 | Finland | 0.561 | 0.586 | 0.595 | 0.589 | 0.591 | 0.554 | 0.566 | 0.7 | MH | MH | MH | MH | MH | MH | MH |
| 27 | Sweden | 0.672 | 0.685 | 0.693 | 0.687 | 0.687 | 0.665 | 0.675 | 0.3 | H | H | H | H | H | H | H |
| | min | 0.298 | 0.312 | 0.321 | 0.333 | 0.334 | 0.313 | 0.321 | | L | L | L | L | L | L | L |
| | max | 0.672 | 0.685 | 0.693 | 0.687 | 0.687 | 0.665 | 0.675 | . | H | H | H | H | H | H | H |

Note: $r_i$ – average annual rate of change (%). Level of development: VH – very high, H – high, MH – medium-high, ML – medium-low, L-low, VL-very low.

*Source: Own calculation based on data from Eurostat.*

**Table 2. Values of synthetic measures for the level of standard of living for European Union countries in 2017–2023.**

| i | Country | Synthetic measures | | | | | | | $r_i$ | Levels of development | | | | | | |
|---|---------|------|------|------|------|------|------|------|----|------|------|------|------|------|------|------|
| | | 2017 | 2018 | 2019 | 2020 | 2021 | 2022 | 2023 | | 2017 | 2018 | 2019 | 2020 | 2021 | 2022 | 2023 |
| 1 | Belgium | 0.566 | 0.575 | 0.558 | 0.550 | 0.553 | 0.668 | 0.679 | 1.2 | MH | MH | MH | MH | MH | H | H |
| 2 | Bulgaria | 0.455 | 0.501 | 0.447 | 0.464 | 0.464 | 0.338 | 0.324 | −1.5 | ML | MH | ML | ML | ML | L | L |
| 3 | Czechia | 0.676 | 0.699 | 0.705 | 0.736 | 0.736 | 0.615 | 0.619 | 0.3 | H | H | H | H | H | H | H |
| 4 | Denmark | 0.707 | 0.716 | 0.709 | 0.716 | 0.718 | 0.644 | 0.625 | −0.6 | H | H | H | H | H | H | H |
| 5 | Germany | 0.648 | 0.673 | 0.662 | 0.635 | 0.635 | 0.620 | 0.639 | −0.1 | H | H | H | H | H | H | H |
| 6 | Estonia | 0.516 | 0.577 | 0.581 | 0.589 | 0.591 | 0.459 | 0.466 | 1.1 | MH | MH | MH | MH | MH | ML | ML |
| 7 | Ireland | 0.697 | 0.683 | 0.676 | 0.686 | 0.689 | 0.631 | 0.638 | −0.9 | H | H | H | H | H | H | H |
| 8 | Greece | 0.403 | 0.424 | 0.397 | 0.411 | 0.411 | 0.344 | 0.372 | −0.6 | ML | ML | L | ML | ML | L | L |
| 9 | Spain | 0.622 | 0.679 | 0.587 | 0.601 | 0.602 | 0.534 | 0.529 | −1.2 | H | H | MH | H | H | MH | MH |
| 10 | France | 0.592 | 0.603 | 0.561 | 0.602 | 0.608 | 0.516 | 0.521 | −0.8 | MH | H | MH | H | H | MH | MH |
| 11 | Croatia | 0.437 | 0.490 | 0.476 | 0.495 | 0.500 | 0.443 | 0.477 | 2.0 | ML | ML | ML | ML | ML | ML | ML |
| 12 | Italy | 0.651 | 0.690 | 0.621 | 0.655 | 0.657 | 0.590 | 0.617 | −0.4 | H | H | H | H | H | MH | H |
| 13 | Cyprus | 0.583 | 0.608 | 0.643 | 0.637 | 0.640 | 0.590 | 0.614 | 1.4 | MH | H | H | H | H | MH | H |
| 14 | Latvia | 0.313 | 0.405 | 0.393 | 0.399 | 0.409 | 0.299 | 0.286 | 3.3 | L | ML | L | L | ML | L | L |
| 15 | Lithuania | 0.389 | 0.472 | 0.465 | 0.455 | 0.464 | 0.385 | 0.344 | 2.2 | L | ML | ML | ML | ML | L | L |
| 16 | Luxembourg | 0.771 | 0.806 | 0.803 | 0.814 | 0.810 | 0.782 | 0.789 | 0.8 | H | VH | VH | VH | VH | H | H |
| 17 | Hungary | 0.432 | 0.467 | 0.440 | 0.469 | 0.472 | 0.417 | 0.460 | 1.1 | ML | ML | ML | ML | ML | ML | ML |
| 18 | Malta | 0.748 | 0.738 | 0.733 | 0.755 | 0.754 | 0.717 | 0.730 | −0.3 | H | H | H | H | H | H | H |
| 19 | Netherlands | 0.775 | 0.784 | 0.775 | 0.801 | 0.802 | 0.764 | 0.758 | 0.2 | H | H | H | VH | VH | H | H |
| 20 | Austria | 0.733 | 0.755 | 0.742 | 0.753 | 0.759 | 0.682 | 0.695 | −0.1 | H | H | H | H | H | H | H |
| 21 | Poland | 0.498 | 0.495 | 0.538 | 0.537 | 0.536 | 0.499 | 0.502 | 0.8 | ML | ML | MH | MH | MH | ML | MH |
| 22 | Portugal | 0.527 | 0.584 | 0.552 | 0.533 | 0.530 | 0.475 | 0.499 | 0.1 | MH | MH | MH | MH | MH | ML | ML |
| 23 | Romania | 0.392 | 0.426 | 0.385 | 0.390 | 0.390 | 0.307 | 0.308 | −1.3 | L | ML | L | L | L | L | L |
| 24 | Slovenia | 0.586 | 0.631 | 0.627 | 0.634 | 0.638 | 0.572 | 0.585 | 1.0 | MH | H | H | H | H | MH | MH |
| 25 | Slovakia | 0.550 | 0.503 | 0.542 | 0.542 | 0.535 | 0.417 | 0.469 | −2.0 | MH | MH | MH | MH | MH | ML | ML |
| 26 | Finland | 0.613 | 0.627 | 0.620 | 0.636 | 0.640 | 0.550 | 0.536 | −0.4 | H | H | H | H | H | MH | MH |
| 27 | Sweden | 0.777 | 0.793 | 0.762 | 0.779 | 0.783 | 0.705 | 0.707 | −0.6 | H | H | H | H | H | H | H |
| | min | 0.313 | 0.405 | 0.385 | 0.390 | 0.390 | 0.299 | 0.286 | | L | ML | L | L | L | L | L |
| | max | 0.777 | 0.806 | 0.803 | 0.814 | 0.810 | 0.782 | 0.789 | | H | VH | VH | VH | VH | H | H |

Note: $r_i$ – average annual rate of change (%). Level of development: VH – very high, H – high, MH – medium-high, ML – medium-low, L – low, VL – very low.

*Source: Own calculation based on data from Eurostat.*

Six countries – Bulgaria, Greece, Croatia, Romania, Hungary, Latvia – remained in a weak position during the entire period. The most notable improvements in development position from 2017 to 2023 were observed in Hungary, Latvia, Lithuania, Czechia, and Croatia (Fig 4). By contrast, minimal changes were noted in German, Finland, France, Denmark, and Sweden. A noteworthy trend emerged: countries with weaker initial development positions generally recorded stronger progress over time, while countries with stronger development positions exhibited less pronounced improvements.

This analysis underscores the disparities within the European Union in terms of both sustainable development and living standards. Within the European Union, there are significant differences between countries and regions regarding sustainable development and living standards. Countries like Sweden, Denmark, Luxembourg, Austria, and the Netherlands achieve high levels of sustainable development, which means they have well-developed infrastructure, high education

**Table 3. The type of development positions are determined by the level of sustainable development and standard of living and corresponding development scenarios for European countries in 2017.**

| Names of position | Number of countries | Countries |
|---|---|---|
| Strong effective position | 11 | **Sweden**, **Austria**, **Luxembourg**, **Denmark**, **Slovenia**, **Czechia**, **Finland**, **Netherlands**, **Ireland**, **Estonia**, **France** |
| Preemptive position | 3 | **Lithuania**, Slovakia, Poland |
| Moderate position | 6 | **Germany**, **Spain**, **Italy,** Belgium, Cyprus, Malta |
| Weak position | 7 | **Bulgaria**, **Greece**, **Croatia**, **Romania**, **Hungary**, **Latvia**, Portugal |

Note: Bold indicates countries with the same position in 2023.

*Source: Own elaboration based on* Table 1.

**Table 4. The type of development positions by the level of sustainable development and standard of living and corresponding development scenarios for European countries in 2023.**

| Names of position | Number of countries | Countries |
|---|---|---|
| Strong effective position | 17 | **Czechia**, **Denmark**, **Estonia**, **Ireland**, **Luxembourg**, **Netherlands**, **Austria**, **Slovenia**, **Finland**, **Sweden**, **France,** Slovakia, Portugal, Poland, Malta, Belgium, Cyprus |
| Preemptive position | 1 | **Lithuania** |
| Moderate position | 3 | **Germany**, **Spain**, **Italy** |
| Weak position | 6 | **Bulgaria**, **Greece**, **Croatia**, **Romania**, **Hungary**, **Latvia** |

Note: Bold indicates countries with the same position in 2017.

*Source: Own elaboration based on* Table 2.

levels, advanced healthcare systems, and strong economies. In contrast, countries in Eastern and Southern Europe, such as Bulgaria, Romania, and Greece, often face higher unemployment rates, lower incomes, and poorer access to public services.

To reduce these disparities, the European Union implements various policies and programs aimed at supporting less developed regions. This includes investments in infrastructure, education, research and development, and support for small and medium-sized enterprises. In countries with lower development levels, the EU focuses on improving basic public services, increasing access to education and vocational training, and promoting sustainable economic development.

Highly developed countries continue to invest in innovation and research to maintain their competitiveness in the global market. They also focus on transitioning to a green economy, which includes investments in renewable energy, sustainable transportation, and environmentally friendly technologies. The analysis of the development trajectories of European countries from 2017 to 2023, based on social, economic, environmental, and quality of life variables, holds significant importance for both policymakers and researchers. These findings provide critical insights into the progress and challenges faced by individual countries. One of the key conclusions derived from this study is the overall positive development trend observed in most countries. This improvement may be attributed to factors such as investments in infrastructure, the implementation of social programs, and heightened environmental awareness. However, it is important to note that such improvements do not always translate directly into sustainable development. This indicates that, in some cases, critical social, economic, or environmental dimensions may not have been fully addressed, reflecting potential gaps or neglect in balanced development efforts.

A closer examination reveals that the majority of countries have shifted upward on the sustainable development matrix, signalling a positive trajectory in their development. That said, there are notable exceptions. Lithuania, for example, deviated from the overall trend with a decline in its position, and while the Czech Republic did not shift on the matrix, it

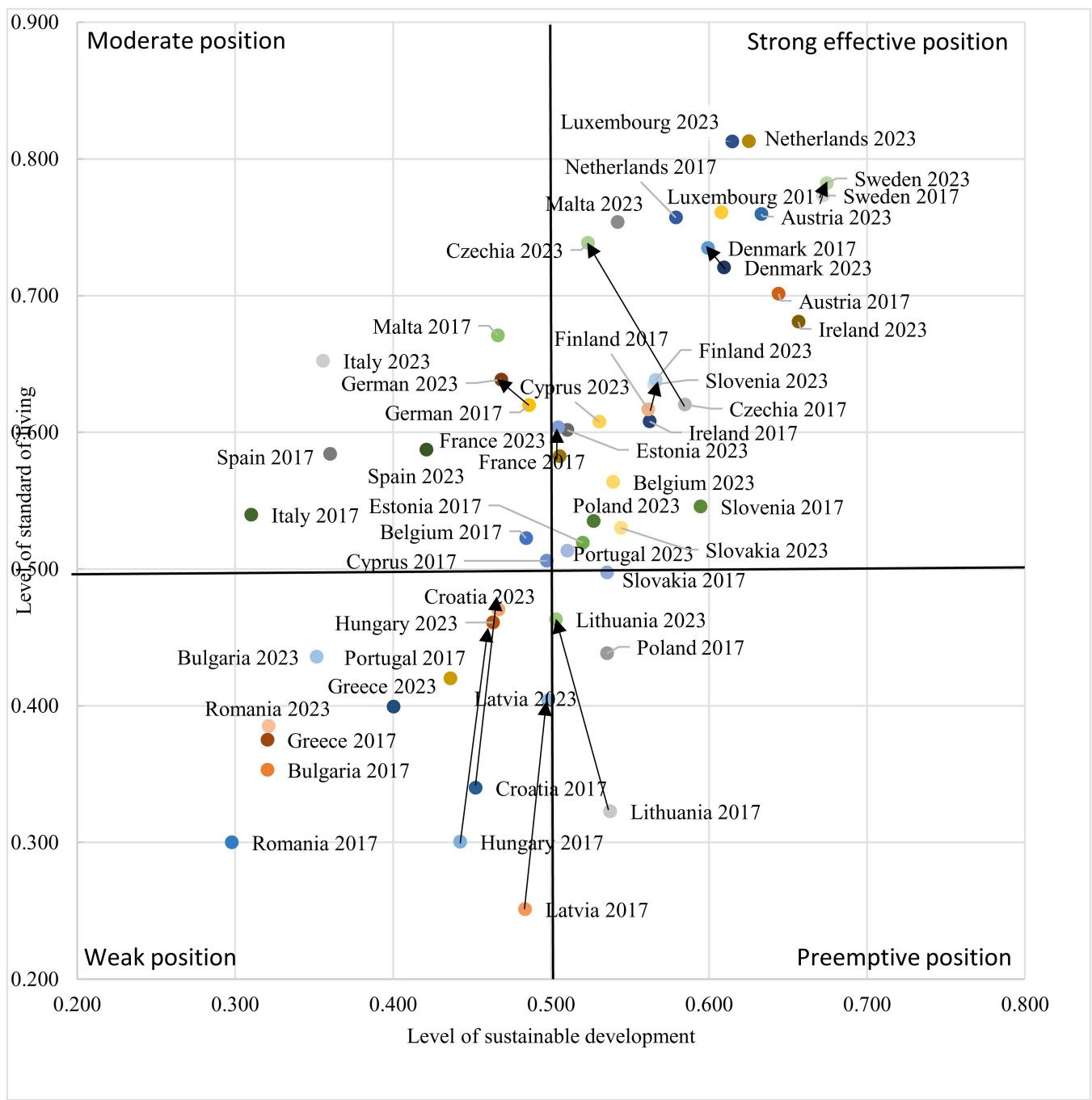

**Fig 4. Development position of European Union countries in 2017 and 2023.** Note: The five largest and five smallest changes have been high-lighted. The arrows indicate the countries that experienced the largest and smallest changes from 2017 to 2023. *Source: Own calculation based on data from Eurostat.*

experienced a minor decline in its sustainable development performance. These exceptions raise important questions for further research, particularly concerning the causes behind such dips in sustainable development over the seven years. Evaluating the circumstances faced by Lithuania and the Czech Republic in comparison to outperforming countries may offer valuable insights into corrective measures and development strategies better aligned with sustainability objectives. The COVID-19 pandemic, which lasted from 2020 to 2023, had a significant impact on the world. During this time, many countries faced health, social, and economic challenges that affected their sustainable development. Economically, the COVID-19 pandemic triggered the largest global economic crisis in over a century. Many businesses worldwide were forced to close, leading to a sharp rise in unemployment rates. The global Gross Domestic Product (GDP) fell significantly, having far-reaching consequences for economies around the world. However, the economic impact of the pandemic was uneven, particularly affecting emerging economies that often-lacked sufficient resources to effectively counter the crisis. The pandemic also posed significant challenges to achieving the Sustainable Development Goals (SDGs). Progress in key areas such as poverty reduction, health, and education were set back. Many people were pushed into extreme poverty, reversing years of efforts to improve living conditions globally. The pandemic also highlighted existing social and economic inequalities, further complicating the realization of the SDGs.

To deepen the understanding of these relationships, two logit models were estimated to identify which factors of living standards most strongly influence sustainable development, and vice versa (see Tables 5 and 6). The empirical analysis revealed that variables such as the share of inactive population in the total population and severe material and social deprivation rate had the most substantial negative impacts on the likelihood of countries achieving higher levels of sustainable development, with odds ratios of 0.789 and 0.894, respectively (Table 5). This indicates that a higher share of inactive population and higher severe material and social deprivation rate significantly reduce the probability of improving the level of sustainable development (21,1% and 10,6%, respectively). Additionally, material and social deprivation rate

**Table 5. Odds ratios from an unordered logit model of sustainable development in European Union countries.**

| $y_j$ | Explanatory variables | Odds ratios | $p$-value | Significance level indicators |
|---|---|---|---|---|
| $y_1$ | the mean income of the population aged 18 and over | 1.000094986 | <0.0001 | *** |
| $y_2$ | material and social deprivation rate | 0.939960900 | 0.0057 | *** |
| $y_3$ | severe material and social deprivation rate | 0.894131878 | 0.0140 | ** |
| $y_6$ | inability to make ends meet | 0.982773109 | 0.0318 | ** |
| $y_7$ | share of inactive population in the total population | 0.789038629 | <0.0001 | *** |

**$p$-value<0.01, indicating that there is less than a 1% chance that the result is due to random variation.

***p-value<0.001, indicating that there is less than a 0.1% chance that the result is due to random variation. *Source: Own calculations using the Gretl package.*

**Table 6. Odds ratios from an unordered logit model of the standard of living in European Union countries.**

| $x_j$ | Explanatory variables | Odds ratios | $p$-value | Significance level indicators |
|---|---|---|---|---|
| $x_1$ | people at risk of poverty or social exclusion | 0.863914045 | <0.0001 | *** |
| $x_5$ | proportion of population aged 65 and over | 0.655798960 | <0.0001 | *** |
| $x_{11}$ | final energy consumption in households per capita | 0.997435654 | <0.0001 | *** |
| $x_{12}$ | recycling rate of municipal waste | 1.046612230 | <0.0001 | *** |
| $x_{13}$ | average $CO_2$ emissions per km from new passenger cars | 0.914196264 | <0.0001 | *** |
| $x_{14}$ | share of renewable energy in gross final energy consumption | 0.932147980 | <0.0001 | *** |
| $x_{15}$ | share of area under organic farming in the total utilized agricultural area | 1.161987615 | <0.0001 | *** |

***p-value<0.001, indicating that there is less than a 0.1% chance that the result is due to random variation.

*Source: Own calculations using the Gretl package.*

and the inability to make ends meet also negatively impacted sustainable development. The material and social deprivation rate is one of the key factors influencing sustainable development in European Union countries. The odds ratio for this variable is 0.940, indicating that a higher material and social deprivation rate is associated with a lower probability of achieving a higher level of sustainable development (by 6%). A high material and social deprivation rate suggest that a significant portion of the population lacks access to basic goods and services, negatively impacting quality of life and development opportunities. In the context of sustainable development, a high level of deprivation can limit a society's ability to invest in education, health, and other key areas essential for long-term development. The inability to make ends meet refers to the financial difficulty experienced by individuals or households when their income is insufficient to cover their basic living expenses. This can include essential costs such as housing, food, utilities, healthcare, and other necessary expenditures. The inability to make ends meet is a significant factor negatively impacting sustainable development in European Union countries. The odds ratio for this variable is 0.983, indicating that higher levels of financial difficulty are associated with a lower probability of achieving a higher level of sustainable development (by 1,7%). This suggests that when a larger portion of the population struggles to cover their basic needs, it can hinder overall progress towards sustainable development. On the other hand, factor like the mean income of the population aged 18 and over showed a small yet positive influence, suggesting that higher mean income slightly increases the possibilities of achieving a higher level of sustainable development.

In summary, while higher mean income positively influences sustainable development, factors such as a higher share of inactive population, higher severe material and social deprivation rate, and higher rates of deprivation and financial difficulty hinder progress towards sustainable development.

The empirical analysis revealed that variables such as the proportion of population aged 65 and over and people at risk of poverty or social exclusion had the most substantial negative impacts on the standard of living, with odds ratios of 0.656 and 0.864, respectively. This indicates that a higher proportion of elderly population and a higher risk of poverty or social exclusion significantly reduce the probability of achieving a higher standard of living (by 34.4% and 13.6%, respectively). On the other hand, factors like the share of area under organic farming in the total utilized agricultural area and recycling rate of municipal waste showed a positive influence, with odds ratios of 1.162 and 1.047, respectively, suggesting that higher organic farming and recycling rates slightly increase the standard of living (by 16.2% and 4.7%, respectively). Additionally, the final energy consumption in households per capita and share of renewable energy in gross final energy consumption also impacted the standard of living, with odds ratios of 0.997 and 0.932, respectively. However, these factors did not significantly decrease the standard of living. In summary, while higher organic farming and recycling rates positively influence the standard of living, factors such as a higher proportion of elderly population, higher risk of poverty or social exclusion, and lower energy consumption per capita hinder improvements in the standard of living.

Detailed analyses of the interconnected dimensions of sustainable development—social, economic, and environmental—demonstrate their mutual dependencies [106]. For instance, improvements in social indicators, such as reducing poverty rates, not only enhance living standards but also alleviate economic burdens. Similarly, investments in environmental protection can enhance social conditions, including health and quality of life. Nevertheless, it is essential to emphasize that isolated improvements in these indicators do not automatically ensure sustainable development. Sustainable development requires maintaining a balance among social, economic, and ecological priorities, as neglecting any one dimension may result in unintended trade-offs. For example, some countries may achieve short-term economic growth at the expense of environmental degradation or increasing social inequality.

There are numerous studies in the literature that document the impact of various factors on sustainable development. For instance, Mazza [107] emphasizes the importance of integrating economic, social, and environmental aspects to achieve sustainable development. Neglecting any of these aspects can lead to unsustainable outcomes. In our study, we observed that most countries shifted their positions on the sustainable development matrix, indicating a positive development trend.

In conclusion, the analysis shows an overall trend of improvement in the development of EU countries from 2017 to 2023. However, exceptions to this trend highlight the need for further investigation into specific cases. The strong interconnectedness between social, economic, and environmental dimensions underscores the need for a holistic, sustainable approach to development. Policymakers, researchers, and other stakeholders should prioritize ongoing data monitoring and analysis to address development gaps effectively and contribute to a more sustainable and balanced future for the region.

Yamaguchi et al. [108] "observed that much research is still very disciplinary" means that a significant portion of the research on Sustainable Development Goals (SDGs) is conducted within specific academic disciplines or fields. This implies that researchers are often focusing on their own specialized areas without much collaboration or integration with other disciplines. In the context of SDGs, which are inherently multidisciplinary and interconnected, this disciplinary approach can be limiting. It suggests that there is a need for more interdisciplinary research that combines insights and methods from different fields to address the complex and interrelated nature of sustainable development challenges more effectively. Our research, which focuses on sustainable development as a whole and on the quality of life, is valuable because it embraces this interdisciplinary approach. By integrating various perspectives and expertise, we can better understand and tackle the multifaceted issues related to sustainable development and improve overall living standards.

## Conclusions

The primary motivation for this research was to explore the evolving dynamics between sustainable development and the standard of living within the European Union. By employing various methods, such as modified positional TOPSIS and ordered logit models, we aimed to gain a comprehensive understanding of these changes. Our key findings reveal significant improvements in sustainable practices, which have positively impacted the quality of life for EU inhabitants. These insights underscore the practical implications of integrating sustainable development policies, suggesting that continued emphasis on sustainability can lead to enhanced living standards across the region.

The research paper has provided a comprehensive exploration of the intricate relationship between sustainable development and the standard of living within the European Union. The analysis revealed marked differentiation in levels of sustainable development and living standards across EU member states. These disparities underscore the complex and heterogeneous nature of development trajectories within the Union, highlighting significant variations in economic performance, social policies, and environmental practices. This points to the need for targeted interventions that account for the unique challenges and opportunities faced by each country.

A general upward trend was observed during the analysed period from 2017 to 2023, with both sustainable development and living standards improving across the EU. This trend reflects encouraging progress in aligning economic growth with social and environmental priorities. However, despite these advancements, pronounced disparities continue to persist among member states, indicating that further efforts are necessary to achieve greater convergence in development outcomes. The COVID-19 pandemic has had a significant impact on sustainable development and living standards, causing regression in some countries. Due to the economic and social disruptions caused by the pandemic, certain member states have experienced a decline in their development levels. This situation underscores the need for continued efforts to address these disparities and support economic recovery and growth in all regions. To counteract these negative effects, coordinated international actions are necessary. Investments in sustainable development, social protection, the green economy, and digitalization can accelerate economic recovery and help achieve sustainable development goals.

The study confirmed a strong relationship between a population's standard of living and the sustainable development levels of their country. This relationship reinforces the essential interconnectedness of economic prosperity, social well-being, and environmental sustainability. It also highlights the need for a balanced approach that ensures improvements in one dimension do not come at the expense of another.

The formulated hypotheses were positively verified based on the analysis conducted in our research. The analysis confirmed that the levels of sustainable development and living standards in European countries from 2017 to 2023 showed significant differences. Northern European countries consistently achieved higher indicators in both areas compared to Southern and Eastern European countries (Hypothesis 1). The developmental positions of European countries with high levels of sustainable development and living standards were found to be stable during the studied period, with only minor changes in the rankings (Hypothesis 2). Moreover, the research identified significant changes in the developmental positions of European countries with low levels of sustainable development and living standards during the studied period (Hypothesis 3).

Logit models allowed for the verification of Hypotheses 4 and 5. The findings demonstrated that higher average income positively influences sustainable development, while deprivation and financial difficulties hinder progress towards sustainable development (Hypothesis 4). The study confirmed that higher rates of organic farming and recycling positively influence the standard of living. Conversely, a higher risk of poverty and social exclusion were found to hinder improvements in the standard of living (Hypothesis 5).

The application of the hybrid multi-criteria decision-making procedure, specifically MP-TOPSIS, proved highly effective in assessing development positions of EU countries. This methodological approach facilitated a nuanced understanding of the intricate interplay between sustainable development and quality of life. It allowed for the systematic evaluation of multiple dimensions and provided meaningful insights that can inform both academic research and policymaking.

These findings carry important implications for policymakers, researchers, and practitioners across the EU. The demonstrated differentiation across countries emphasizes the need for tailored, multi-dimensional policy frameworks that address specific social, economic, and environmental challenges. The positive trend of progress observed in sustainable development and living standards highlights the effectiveness of ongoing EU initiatives but also calls for a redoubling of efforts to bridge gaps between lagging and high-performing countries. The confirmed intersection between sustainable development and living standards underscores the importance of integrated development strategies. Policymakers must ensure that initiatives targeting economic or environmental growth are equally matched by efforts to enhance social well-being and equity across populations. Finally, this research underscores the utility of methodologies such as MP-TOPSIS in evaluating and addressing complex development challenges. This approach not only provides a sophisticated analytical framework but also lays the foundation for future studies to build upon, particularly in exploring causal relationships and long-term impacts of development policies in the EU.

This study contributes to the broader discourse on sustainable development by offering actionable insights into the interplay between sustainability and quality of life. It provides a valuable basis for shaping policies and research that promote equitable, inclusive, and sustainable development across the European Union.

## Implications and recommendations

The findings of this research offer numerous important implications for policymakers, researchers, and practitioners working on sustainable development and improving living standards. Firstly, pronounced differentiation among EU countries emphasizes the necessity for tailored policy interventions. These policies should address the unique development challenges and opportunities of individual member states while reflecting their specific social, economic, and environmental contexts. At the same time, coordinated efforts at the EU level can help harmonize national policies, fostering alignment with shared sustainability goals and long-term regional strategies.

Secondly, the interconnectedness of sustainable development and living standards highlights the need for an integrated approach that balances economic growth, social inclusion, and environmental preservation. Policymakers should prioritize strategies that simultaneously enhance these dimensions, ensuring that progress in one area complements and supports advancements in the others. Such balanced policies are critical to improving citizens' quality of life while advancing sustainability objectives.

Thirdly, the positive trends observed in the study reinforce the importance of regular monitoring and analysis to track progress and identify emerging challenges. Monitoring can reveal potential disparities or areas where policies require recalibration. Conducting further research can also provide deeper insights into the drivers of sustainable development, enabling evidence-based decision-making and the development of adaptive strategies that respond to evolving needs.

Fourthly, the demonstrated success of MP-TOPSIS in this study underscores its potential for broader application in analysing complex development phenomena. Researchers and practitioners may find value in exploring and adapting this methodology for other contexts, enabling them to evaluate multi-dimensional indicators with greater precision. Such innovations can support informed management of development projects and strategic planning across a variety of domains.

This research has highlighted critical changes in the development positions of EU countries between 2017 and 2023 by evaluating levels of sustainable development and standard of living. It demonstrates how sustainability and living standards interact dynamically, providing a nuanced understanding of the European development landscape. Importantly, the conclusions of this study serve as both an analytical tool for assessing developmental progress and a practical guide for informing policy formulation and implementation.

As the EU continues to face the complexities of global challenges such as climate change, inequality, technological transformation, and economic volatility, the findings of this research emphasize the need for inclusive, innovative, and foresighted strategies. The results offer a foundation for building more effective and context-sensitive interventions that promote not only economic growth but also social equity and environmental resilience.

By actively integrating these recommendations, the EU can take proactive steps to ensure a prosperous, equitable, and sustainable future for all its inhabitants, fostering a development model that serves as a benchmark for global sustainability efforts.

## Author contributions

**Conceptualization:** Aleksandra Łuczak, Klara Cermakova, Slawomir Kalinowski, Eduard Hromada.

**Data curation:** Aleksandra Łuczak, Oskar Szczygieł.

**Formal analysis:** Aleksandra Łuczak, Klara Cermakova.

**Funding acquisition:** Klara Cermakova.

**Investigation:** Aleksandra Łuczak, Eduard Hromada.

**Methodology:** Aleksandra Łuczak.

**Project administration:** Klara Cermakova, Eduard Hromada.

**Resources:** Eduard Hromada.

**Supervision:** Aleksandra Łuczak, Klara Cermakova.

**Validation:** Slawomir Kalinowski.

**Visualization:** Aleksandra Łuczak, Slawomir Kalinowski, Oskar Szczygieł.

**Writing – original draft:** Aleksandra Łuczak, Klara Cermakova.

**Writing – review & editing:** Aleksandra Łuczak, Klara Cermakova, Slawomir Kalinowski, Eduard Hromada.

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
