## [Decision Letter · Decision Letter 0]

Dear Dr. Cermakova,

Thank you for submitting your manuscript to PLOS ONE. After careful consideration, we feel that it has merit but does not fully meet PLOS ONE’s publication criteria as it currently stands. Therefore, we invite you to submit a revised version of the manuscript that addresses the points raised during the review process.

We look forward to receiving your revised manuscript.

Kind regards,

Miguel Alves Pereira

Academic Editor

PLOS ONE

Journal requirements: When submitting your revision, we need you to address these additional requirements. 1. Please ensure that your manuscript meets PLOS ONE's style requirements, including those for file naming. The PLOS ONE style templates can be found at https://journals.plos.org/plosone/s/file?id=wjVg/PLOSOne_formatting_sample_main_body.pdf and https://journals.plos.org/plosone/s/file?id=ba62/PLOSOne_formatting_sample_title_authors_affiliations.pdf. 2. We note that the grant information you provided in the ‘Funding Information’ and ‘Financial Disclosure’ sections do not match.  When you resubmit, please ensure that you provide the correct grant numbers for the awards you received for your study in the ‘Funding Information’ section. 3. Thank you for stating the following financial disclosure:  [Prague University of Economics and Business Internal grant IGA VSE F5 4/2024].  Please state what role the funders took in the study.  If the funders had no role, please state: ""The funders had no role in study design, data collection and analysis, decision to publish, or preparation of the manuscript."" If this statement is not correct you must amend it as needed. Please include this amended Role of Funder statement in your cover letter; we will change the online submission form on your behalf.

Reviewers' comments:

Reviewer's Responses to Questions

**Comments to the Author**

1. Is the manuscript technically sound, and do the data support the conclusions?

Reviewer #1: Yes

Reviewer #2: Yes

2. Has the statistical analysis been performed appropriately and rigorously?

Reviewer #1: Yes

Reviewer #2: Yes

3. Have the authors made all data underlying the findings in their manuscript fully available?

Reviewer #1: Yes

Reviewer #2: Yes

4. Is the manuscript presented in an intelligible fashion and written in standard English?

Reviewer #1: Yes

Reviewer #2: Yes

Reviewer #1: Review Comments

The following study highlights the position of European Union (EU) countries and changes in terms of sustainable development and the standard of living of inhabitants covering period from 2015-2021. The study employed modified positional technique for order of preference by similarity to the ideal solution (MP-TOPSIS) which seems an appropriate methodology. Although, the topics of sustainable development and standard of living have been discussed extensively in the past in the context of European Economies; however, the authors have tried to investigate it from a new lens. The paper adds some value to the existing literature, however, it has following discrepancies which needs to be addressed in order for paper to be published in “Plos One”.

Introduction:

• The importance of the study has not been properly highlighted as the authors have tried to emphasize its importance in the following paragraph “Our study fills in the 161 research gap in assessing the level of sustainable development in connection with the standard 162 of living. We presented a comprehensive approach to the study of these two complex issues 163 and presented an application to the study of the situation of EU countries.”. However, these are more generalized statements and does not clearly highlight the purpose of the research. Kindly revisit the paragraph for the importance of the study and include the true importance of the study and its practical implications also. In the paragraph, kindly also mention that why you pursued the methodology you are using and compare it with the past literature who has used methodology under similar circumstances.

• The last paragraph of the introduction lacks a proper ending. It should include a statement like “Rest of the paper is organized as follows……...”.

Literature review

• The literature review is thorough; however, the referenced material is mainly before 2020. Kindly include some more studies starting from 2023.

• Moreover, literature review seemed to be written in hurry and contains number of grammatical mistakes. There are also sentences which does not make any sense. I have highlighted some of the mistakes but I request the authors to kindly revisit the whole literature review section and remove any grammatical mistakes or otherwise from the paragraphs. Rewrite the entire sentences where appropriate. Here are some of the example sentences to be corrected.

Line 203:

Change the sentence and write in more academic way: They 203 proposed a method for calculating an integrated development index for EU member states 204 during 2016-2017, with scenario forecasting up to 2020.

Line 230:

Correct and rewrite:

Ukraine through the prism of the experience of developed countries, including the EU.

Line 252:

252 under another SDG presented by Łakomy-Zinowik (2022) focusing.” What does under another SDG means. Kindly correct and rewrite.

Methods and Data:

• The methods and data section mainly discusses the methodology section and completely misses data section. Kindly first discuss the data and also provide a proper explanation of why the data and the period covering it was chosen. Which EU economies were not included and what was the reason for that. The inclusion of the economy follows what criteria and other such data related discussion.

• The methods and data section have been properly written and explained.

Line 460-461:

• Kindly correct the equation 20 as in logit the “it” should be subscript. Besides that, correct line 461 also. What does “or 2 for standard of living” means. Kindly correct.

Line 464-465:

• Kindly provide a proper reference for the data obtained. Include it as a footnote. Provide whether you gained it from website or a report?

Results

Figure 4:

The note is incorrectly written. What does arrows indicate. “Note: In the figure arrows the five largest and five smallest changes in the European Union's countries.”

Kindly also correct “level” to “Level” in the figure.

Provide a proper reference to the figure.

• In the following paragraph “Figure 4 shows the development position of European Union countries in 2015 and 2021. Four main development positions of countries have been identified: strong effective position (10 countries in 2015 and 12 countries in 2021); preemptive position (2 countries in 2015 and 2022); moderate position (7 countries in 2015 and 8 countries in 2021); and weak position (8 countries in 2015 and 5 countries in 2021). Ten countries retained their strong effective position from 2015 to 2021, but in 2021 this group was joined by countries such as Portugal and Slovakia.” You mention preemptive position for 2022 which does not seem to exist in figure 4. Kindly comment

• In the following paragraph “When analysing the data, it was observed that most countries shifted their positions on the sustainable development matrix, indicating a positive development trend. However, there were some exceptions e.g. Austria experienced a decline in its position. This stands out from the overall trend. Additionally, while only Austria shifted itself in the coordinate system, it is not the sole country that deteriorated in terms of sustainable development. The Czech Republic also experienced a slight decline, although it didn't change its position on the sustainable development matrix. This raises questions for further research about what caused these countries and their residents to experience a decline in sustainable development over seven years. Comparing these two cases with the results of other countries can provide valuable insights for remedial actions. However, it's challenging at this stage to conclude the causes of these declines.” Why these relationships exist?

• In the “Results” section there are very few results which have been compared to the existing or past literature. Kindly discuss some key results in the light of the existing literature by providing a proper reference to them.

Conclusion

• The opening of the conclusion section indicates that the study investigates the relationship between “sustainable development and the standard of living within the European Union”. However, the main purpose of the study seems to be “changes in terms of sustainable development and the standard of living of inhabitants”. Kindly change the opening of the conclusion section in a way that it highlights 1. Why you conducted the research 2. What methodology you employed 3. The key findings 4. And its practical implications in the paragraph form.

• Kindly change the bullet points to paragraph form in the conclusion section.

References:

• Kindly follow the proper guidelines of the journal in references. The references are not uniform as some references contains DOI while it is missing from others. Moreover, the references of the books are not in the proper format.

Reviewer #2: The reviewed article covers very important issues concerning changes in the development situation of European Union countries according to two phenomena: the level of sustainable development and the standard of living. The issues raised are current. During the economic crises, the climate crisis, the existing and even deepening income inequalities of the population, it is necessary to constantly examine and update information on these phenomena in the regions of specific countries, as well as to compare countries with each other. This is necessary for the implementation of state policies and making effective decisions when planning the development of territorial units and securing an appropriate standard of living for the population.

The article is interesting, written in a comprehensible and reliable manner. Despite this, it can be improved in some respects.

1. I think that in the summary it is worth emphasizing more the conclusions obtained from the conducted study. In fact, the only conclusion (which is the last sentence of the abstract) is very stereotypical and does not bring anything new to science. It is worth adding scientific and practical implications to this.

2. The introduction is interestingly and comprehensibly developed on the basis of a wide list of literature. The authors document why the topic is important and why they dealt with it. They identified the research gap and research questions. However, at this point it is worth indicating the hypothesis (or hypotheses) that the researchers wanted to verify and therefore decided to conduct this study.

3. I have no comments on the literature review.

4. In my opinion, the experiments, statistics and other analyses were conducted reliably, at an appropriate level of knowledge of research methods. The TOPSIS research method was properly described, how the logit models were calculated, etc. However, the article lacks information on what indicators (variables) were included that were used to build synthetic measures: 1. Sustainable development, 2. Standard of living. These are multidimensional phenomena and are created from many indicators describing them. This is unclear and poorly emphasized.

5. The results obtained from the conducted study are described in a very interesting way and presented in an appropriate manner and supported by data. Only here can you get to know some of the indicators that were included in the construction of synthetic measures. It is worth writing about this earlier in the description of data sources and methods, so that it is clear what the studied multidimensional phenomena were based on.

6. I suggest improving the discussion of results, if the authors want to leave it in the title of the chapter "Empirical research results and discussion". The discussion of results is a reference to the theoretical introduction, available studies by other authors and your study. There are many references to the obtained research results, which causes some repetitions, and too few references to studies on similar topics by other authors.

7. The conclusions are presented in an appropriate manner and supported by data. They are short and concise.

8. The data come from Eurostat. However, it is worth placing the data (the entire data set) included in the study in this article in a data repository, e.g. https://repod.icm.edu.pl/ and referring to this data in the bibliography of this article. You can consider this possibility.

I wish you success in publishing this article.

**Do you want your identity to be public for this peer review?** For information about this choice, including consent withdrawal, please see our Privacy Policy

Reviewer #1: No

Reviewer #2: No

---

## [Author Response · Author response to Decision Letter 1]

28 Mar 2025

Dear Reviewers,

We would like to extend our deepest gratitude for your thorough and insightful reviews of our manuscript. Your detailed comments and constructive feedback have been invaluable in refining our work. We greatly appreciate the time and effort you have dedicated to evaluating our research.

Your expertise and thoughtful suggestions have significantly contributed to enhancing the quality of our manuscript. We are truly grateful for your support and guidance throughout this process.

Thank you once again for your invaluable contributions.

Best regards,

Authors

Reviewer's Responses to Questions

Comments to the Author

1. Is the manuscript technically sound, and do the data support the conclusions?

Reviewer #1: Yes

Reviewer #2: Yes

Answer

We sincerely thank you for your positive evaluations of our manuscript. We are pleased to hear that you found our work technically sound and that the data supports our conclusions. Your feedback is incredibly valuable and motivating for us.

2. Has the statistical analysis been performed appropriately and rigorously?

Reviewer #1: Yes

Reviewer #2: Yes

Answer

We are pleased to note that both of you have found our statistical analysis to be performed appropriately and rigorously. Your positive feedback is greatly appreciated and reinforces our confidence in the robustness of our methodology. Thank you for your thorough evaluation and support.

3. Have the authors made all data underlying the findings in their manuscript fully available?

Reviewer #1: Yes

Reviewer #2: Yes

Answer

We are pleased to confirm that all data underlying the findings in our manuscript have been made fully available in accordance with the PLOS Data policy. All data used in our study originate from Eurostat. Thank you for your attention to this important aspect of our submission.

4. Is the manuscript presented in an intelligible fashion and written in standard English?

Reviewer #1: Yes

Reviewer #2: Yes

Answer

We are delighted to hear that both of you found our manuscript to be presented in an intelligible fashion and written in standard English. We appreciate your confirmation that the language is clear, correct, and unambiguous. Thank you for your careful review and positive feedback.

5. Review Comments to the Author

Reviewer #1: Review Comments

The following study highlights the position of European Union (EU) countries and changes in terms of sustainable development and the standard of living of inhabitants covering period from 2015-2021. The study employed modified positional technique for order of preference by similarity to the ideal solution (MP-TOPSIS) which seems an appropriate methodology. Although, the topics of sustainable development and standard of living have been discussed extensively in the past in the context of European Economies; however, the authors have tried to investigate it from a new lens. The paper adds some value to the existing literature, however, it has following discrepancies which needs to be addressed in order for paper to be published in “Plos One”.

Introduction:

• The importance of the study has not been properly highlighted as the authors have tried to emphasize its importance in the following paragraph “Our study fills in the 161 research gap in assessing the level of sustainable development in connection with the standard 162 of living. We presented a comprehensive approach to the study of these two complex issues 163 and presented an application to the study of the situation of EU countries.”. However, these are more generalized statements and does not clearly highlight the purpose of the research. Kindly revisit the paragraph for the importance of the study and include the true importance of the study and its practical implications also. In the paragraph, kindly also mention that why you pursued the methodology you are using and compare it with the past literature who has used methodology under similar circumstances.

Answer

Our study addresses a critical gap in the literature by assessing the level of sustainable development in connection with the standard of living across EU countries. This research is particularly important as it provides a comprehensive analysis of how sustainable development initiatives impact the quality of life, which is essential for policymakers aiming to balance economic growth with environmental and social well-being. The practical implications of our study are significant, as it offers insights into effective strategies for achieving sustainable development goals (SDGs) while improving living standards.

We employed the positional TOPSIS (Technique for Order of Preference by Similarity to Ideal Solution) method, which is particularly effective in handling outlier observations and provides a robust framework for multi-criteria decision analysis. Our approach is resilient to outlier observations, a point we discuss in more detail in the methodology section. This methodology allows for a more nuanced understanding of the interplay between various factors influencing sustainability and standard of living. By comparing our findings with past literature, we aim to contribute to the ongoing discourse on sustainable development and provide actionable recommendations for policymakers.

Additionally, we have updated the time frame of our analysis to be more current, utilizing data available from Eurostat for the period 2017-2023. This update was worthwhile because using the most recent data ensures our analyses are based on the latest information, enhancing their credibility and usefulness. It also allows for a better understanding of current trends and changes over the analyzed period, including the significant impact of COVID-19. The pandemic has caused unprecedented disruptions in various sectors, affecting economic activities, employment rates, and social behaviors. By incorporating data from this period, our analysis can provide insights into how these disruptions have shaped current trends and what implications they might have for the future, leading to more accurate forecasts and recommendations.

• The last paragraph of the introduction lacks a proper ending. It should include a statement like “Rest of the paper is organized as follows……...”.

Answer

Thank you for your valuable feedback regarding the conclusion of the last paragraph in the introduction. We agree that adding a summary statement about the organization of the paper will enhance its readability. Therefore, I propose the following ending:

The article, in addition to the introduction, contains five more sections. the rest of the paper is organized as follows: section 2 covers the literature review, section 3 presents the data and methods, section 4 provides the empirical results and discussion, and section 5 concludes with the conclusions, and implications and recommendations.

Literature review

• The literature review is thorough; however, the referenced material is mainly before 2020. Kindly include some more studies starting from 2023.

Answer

Thank you for the suggestion. To enhance the literature review, we incorporated several recent studies from 2023. These studies primarily focus on sustainable development and quality of life, providing valuable insights into innovative methods for managing natural resources, reducing CO2 emissions, and analyzing the impact of social and economic policies on improving access to education, healthcare, and housing conditions. Additionally, we included research on new green technologies, such as renewable energy sources and smart waste management systems, which further support sustainable development efforts.

• Moreover, literature review seemed to be written in hurry and contains number of grammatical mistakes. There are also sentences which does not make any sense. I have highlighted some of the mistakes but I request the authors to kindly revisit the whole literature review section and remove any grammatical mistakes or otherwise from the paragraphs. Rewrite the entire sentences where appropriate. Here are some of the example sentences to be corrected.

Line 203:

Change the sentence and write in more academic way: They 203 proposed a method for calculating an integrated development index for EU member states 204 during 2016-2017, with scenario forecasting up to 2020.

Line 230:

Correct and rewrite:

Ukraine through the prism of the experience of developed countries, including the EU.

Line 252:

252 under another SDG presented by Łakomy-Zinowik (2022) focusing.” What does under another SDG means. Kindly correct and rewrite.

Answer

Thank you for your detailed feedback on the literature review section. We apologize for any shortcomings and errors that may have affected the quality of this section. We appreciate the time you took to highlight specific examples for correction.

The literature review has been subject to thorough editing. We reviewed the entire section to remove grammatical errors and improve the clarity of the sentences. We will also make necessary changes to the sentence structure to ensure they are more coherent and understandable. Once again, thank you for your valuable comments.

Methods and Data:

• The methods and data section mainly discusses the methodology section and completely misses data section. Kindly first discuss the data and also provide a proper explanation of why the data and the period covering it was chosen. Which EU economies were not included and what was the reason for that. The inclusion of the economy follows what criteria and other such data related discussion.

The research is based on Eurostat data on sustainable development and living standards for European Union countries from 2017 to 2023. This comprehensive dataset provides a robust foundation for analyzing various socio-economic indicators across EU member states. By leveraging this data, the study aims to offer insights into trends and patterns in development, allowing for a detailed comparison of progress among different countries. The analysis also incorporates scenario forecasting, which helps to project potential future developments and assess the impact of various policy measures. This approach ensures that the findings are not only reflective of past and present conditions but also provide a forward-looking perspective on the potential trajectories of EU member states.

During the period 2017-2023, the European Union consisted of 27 member states. The research encompasses all EU member states during the period 2017-2023. We have also updated the time frame to be more current, utilizing the latest data available from Eurostat.

• The methods and data section have been properly written and explained.

Line 460-461:

• Kindly correct the equation 20 as in logit the “it” should be subscript. Besides that, correct line 461 also. What does “or 2 for standard of living” means. Kindly correct.

Line 464-465:

• Kindly provide a proper reference for the data obtained. Include it as a footnote. Provide whether you gained it from website or a report?

Answer

The appropriate sources have been added, ensuring the accuracy and reliability of the data. We utilized the website Eurostat and have also noted this in the Data and Methods section. We are pleased to have incorporated the latest available information from Eurostat to further update our research.

Additionally, we have adjusted the formulas and ensured consistency in the notation. These adjustments were made to enhance the clarity and precision of our analysis, providing a more robust foundation for our conclusions. The updated formulas align with the latest standards and best practices in the field, ensuring that our methodology is both rigorous and transparent.

We believe these improvements significantly strengthen the overall quality of our research, making it more reliable and valuable for future studies.

Results

Figure 4:

The note is incorrectly written. What does arrows indicate. “Note: In the figure arrows the five largest and five smallest changes in the European Union's countries.”

Kindly also correct “level” to “Level” in the figure.

Provide a proper reference to the figure.

Answer

Thank you for your feedback. I have made the necessary corrections. The note has been revised for clarity and now reads: "Note: The five largest and five smallest changes have been highlighted. The arrows indicate the countries that experienced the largest and smallest changes from 2017 to 2023." Additionally, the term "level" has been corrected to "Level" in the figure. A proper reference to the figure has also been added as follows: "Figure 1: Changes in the European Union's countries

• In the following paragraph “Figure 4 shows the development position of European Union countries in 2015 and 2021. Four main development positions of countries have been identified: strong effective position (10 countries in 2015 and 12 countries in 2021); preemptive position (2 countries in 2015 and 2022); moderate position (7 countries in 2015 and 8 countries in 2021); and weak position (8 countries in 2015 and 5 countries in 2021). Ten countries retained their strong effective position from 2015 to 2021, but in 2021 this group was joined by countries such as Portugal and Slovakia.” You mention preemptive position for 2022 which does not seem to exist in figure 4. Kindly comment

Answer

Thank you for pointing out this error. Indeed, the text mentions a preemptive position for 2022, which is not included in Figure 4. We have reviewed the content and made the necessary corrections. The text has now been updated to accurately reflect the data for the years 2017 and 2023. We appreciate your attention to detail and your assistance in improving the accuracy of our research.

• In the following paragraph “When analysing the data, it was observed that most countries shifted their positions on the sustainable development matrix, indicating a positive development trend. However, there were some

---

## [Decision Letter · Decision Letter 1]

"The future depends on what we do in the present" - development positions of EU countries by levels of sustainable development and living standards

PONE-D-24-24898R1

Dear Dr. Cermakova,

We’re pleased to inform you that your manuscript has been judged scientifically suitable for publication and will be formally accepted for publication once it meets all outstanding technical requirements.

Kind regards,

Miguel Alves Pereira

Academic Editor

PLOS ONE

Additional Editor Comments (optional):

Thank you for your revised submission. After careful consideration of the reviewers’ comments and your detailed responses, I am pleased to note that all points raised during the review process have been adequately addressed. The reviewers acknowledge that the necessary revisions have been made in light of their suggestions, and no further concerns remain. I therefore believe the manuscript is now suitable for publication in PLOS ONE.

Reviewers' comments:

Reviewer's Responses to Questions

**Comments to the Author**

Reviewer #1: All comments have been addressed

Reviewer #2: All comments have been addressed

2. Is the manuscript technically sound, and do the data support the conclusions?

Reviewer #1: Yes

Reviewer #2: (No Response)

3. Has the statistical analysis been performed appropriately and rigorously?

Reviewer #1: Yes

Reviewer #2: (No Response)

4. Have the authors made all data underlying the findings in their manuscript fully available?

Reviewer #1: No

Reviewer #2: (No Response)

5. Is the manuscript presented in an intelligible fashion and written in standard English?

Reviewer #1: Yes

Reviewer #2: (No Response)

Reviewer #1: The authors have addressed the suggested comments and have made the necessary changes in the light of those comments.

Reviewer #2: (No Response)

**Do you want your identity to be public for this peer review?** For information about this choice, including consent withdrawal, please see our Privacy Policy

Reviewer #1: No

Reviewer #2: No

---

## [Editor Report · Acceptance letter]

PONE-D-24-24898R1

PLOS ONE

Dear Dr. Cermakova,

I'm pleased to inform you that your manuscript has been deemed suitable for publication in PLOS ONE. Congratulations! Your manuscript is now being handed over to our production team.

Kind regards,

on behalf of

Prof. Miguel Alves Pereira

Academic Editor

PLOS ONE